# The inner mechanics of rhodopsin guanylyl cyclase during cGMP-formation revealed by real-time FTIR spectroscopy

**Paul Fischer[1]\*, Shatanik Mukherjee[2], Enrico Schiewer[1], Matthias Broser[1], Franz Bartl[2], Peter Hegemann[1]**

[1]Institute for Biology, Experimental Biophysics, Humboldt-Universität zu Berlin, Berlin, Germany; [2]Institute of Biology, Biophysical Chemistry, Humboldt University of Berlin, Berlin, Germany

**Abstract** Enzymerhodopsins represent a recently discovered class of rhodopsins which includes histidine kinase rhodopsin, rhodopsin phosphodiesterases, and rhodopsin guanylyl cyclases (RGCs). The regulatory influence of the rhodopsin domain on the enzyme activity is only partially understood and holds the key for a deeper understanding of intra-molecular signaling pathways. Here, we present a UV-Vis and FTIR study about the light-induced dynamics of a RGC from the fungus *Catenaria anguillulae*, which provides insights into the catalytic process. After the spectroscopic characterization of the late rhodopsin photoproducts, we analyzed truncated variants and revealed the involvement of the cytosolic N-terminus in the structural rearrangements upon photo-activation of the protein. We tracked the catalytic reaction of RGC and the free GC domain independently by UV-light induced release of GTP from the photolabile NPE-GTP substrate. Our results show substrate binding to the dark-adapted RGC and GC alike and reveal differences between the constructs attributable to the regulatory influence of the rhodopsin on the conformation of the binding pocket. By monitoring the phosphate rearrangement during cGMP and pyrophosphate formation in light-activated RGC, we were able to confirm the M state as the active state of the protein. The described setup and experimental design enable real-time monitoring of substrate turnover in light-activated enzymes on a molecular scale, thus opening the pathway to a deeper understanding of enzyme activity and protein-protein interactions.

**\*For correspondence:**
paul.fischer.2@hu-berlin.de

**Competing interests:** The authors declare that no competing interests exist.

## Introduction

Microbial rhodopsins (Rhs) are transmembrane proteins that utilize light-induced isomerization of their retinal cofactor to function as light-sensitive ion channels, pumps and sensors. Most recently that list has been complemented by the new class of enzymerhodopsins which comprises Rhs characterized by their light-regulated enzyme function. So far, all members are related to the regulation of the intracellular second messenger cGMP. Throughout most branches of life, the ubiquitous intracellular second messenger molecule cGMP is involved in a wide variety of biological functions such as platelet aggregation, neurotransmission, sexual arousal, gut peristalsis, blood pressure, long bone growth, intestinal fluid secretion, lipolysis, phototransduction, cardiac hypertrophy, and oocyte maturation (*Potter, 2011*). Since enzymerhodopsins enable local control of cGMP and, if modified, cAMP (*Scheib et al., 2018*), they usher in a new direction for optogenetics aiming to unravel the details of these signaling pathways. Thus, a comprehensive understanding of the activation and inactivation of these novel proteins is of utmost interest.

In these photoreceptors, the C-terminus of the rhodopsin is directly linked to either a histidine kinase (histidine kinase rhodopsin), a phosphodiesterase (rhodopsin phosphodiesterase, RPDE), or a guanylyl cyclase (rhodopsin guanylyl cyclase, RGC). The RGC was first discovered in the aquatic

fungus *Blastocladiella emersonii* as the photoreceptor responsible for the phototactic behavior of the fungal zoospores (*Avelar et al., 2014*). In vitro studies confirmed its function as a photoactivated guanylyl cyclase that is inactive in the dark and fully active when illuminated with green light (*Scheib et al., 2015*). More functional details are available from the orthologous protein of the related fungus *Catenaria anguillulae* (*Ca*RGC) including the crystal structure of the constitutively active isolated cyclase domain (*Scheib et al., 2018*; *Butryn et al., 2020*). The results show the photoreceptor to assemble as a homodimer, with the rhodopsin modules responsible to silence the cyclase activity in darkness. Meanwhile, a number of enzymerhodopsins have been purified and enzymatically and spectroscopically characterized (*Trieu et al., 2017*; *Kumar et al., 2017*; *Yoshida et al., 2017*; *Tian et al., 2018*; *Watari et al., 2019*). In all of them, the canonical hepta-helical rhodopsin module is preceded by an additional eighth transmembrane helix (TM0), leading to a cytosolic localization of both termini. To date, however, no full-length structural model of an enzymerhodopsin is available, leaving the mechanism of how photoactivation of the rhodopsin regulates enzyme function, including the role of the cytoplasmic N-terminus, unclear. Recently, the crystal structures of the isolated transmembrane and intracellular catalytic domains were solved separately for RPDEs from choanoflagellates, which are already active in the dark and only moderately stimulated under illumination (*Lamarche et al., 2017*; *Ikuta et al., 2020*). While these structures allow the placement of the additional transmembrane alpha helix for the first time, a detailed description of the structural changes as they occur upon photoactivation in the rhodopsin module and subsequently in the catalytically active sites of the enzyme that lead to substrate turnover is missing so far. Here, we describe a detailed investigation of the RGC from *Catenaria anguillulae* (*Ca*) by using a combined UV-Vis and FTIR spectroscopic analysis. We report on the overall changes of the protein upon photoactivation, mainly occurring in the rhodopsin module, and we identify structural constraints related to substrate binding, catalytic conversion of GTP into cGMP, as well as the product release from the binding site. Our study provides a comprehensive understanding of structure-activity relationships in an enzymerhodopsin.

## Results

### Characterization of photoproducts

An initial spectroscopic characterization of *Ca*RGC identified three subsequent photocycle intermediates designated as K, L, and M-states according to the nomenclature derived from bacteriorhodopsin as prototypical microbial rhodopsin (*Scheib et al., 2018*). The M state decays within a few hundred ms, but the K and L states only rest in the sub-μ to μs time range, which makes them unfeasible for investigation by rapid scan FTIR. Previous studies on Channelrhodopsin 1 and 2 (ChR1, ChR2) have revealed an interaction between an aspartate-cysteine (DC) pair localized close to the retinal C13=C14 bond around which photoisomerization occurs. Disruption of this DC-interaction by mutating the respective residues dramatically slows down the photocycle kinetics (*Ritter et al., 2013*; *Lórenz-Fonfría and Heberle, 2014*; *Nack et al., 2010*). A multiple sequence alignment indicated these residues to be conserved in *Ca*RGC, suggesting that a similar mechanism may be exploited to delay its photocycle. Indeed, as described for ChR1 and ChR2, the C259S mutant of *Ca*RGC decelerated the photocycle about tenfold with seemingly unimpeded functionality.

Since FTIR rapid-scan acquisition rates are limited to tenth of milliseconds, the C259S mutant provides an opportunity to study transient photocycle intermediates of *Ca*RGC and their kinetics with the time resolution available with our new UV-Vis and FTIR spectroscopic setup (Appendix 1).

### UV-Vis

*Figure 1A* shows UV-Vis spectra of the slow cycling C259S mutant acquired at 20°C and −10°C both for single turnover and continuous illumination conditions. Each measurement shows the dark state bleach at around 545 nm (blue). For single turnover conditions at 20°C, the late L state absorbing at 450 nm precedes the M state absorbing at 350 nm. At the same temperature but under continuous illumination, the slower M state accumulates congruously with no traces of K and L state within the limits of accuracy of the measurement. To further retard the photocycle, the measurements were repeated at −10°C. At this temperature, the photostationary equilibrium shows clear populations of K, L, and M species favoring the L state (*Figure 1C*). In case of single turnover, the K state decay

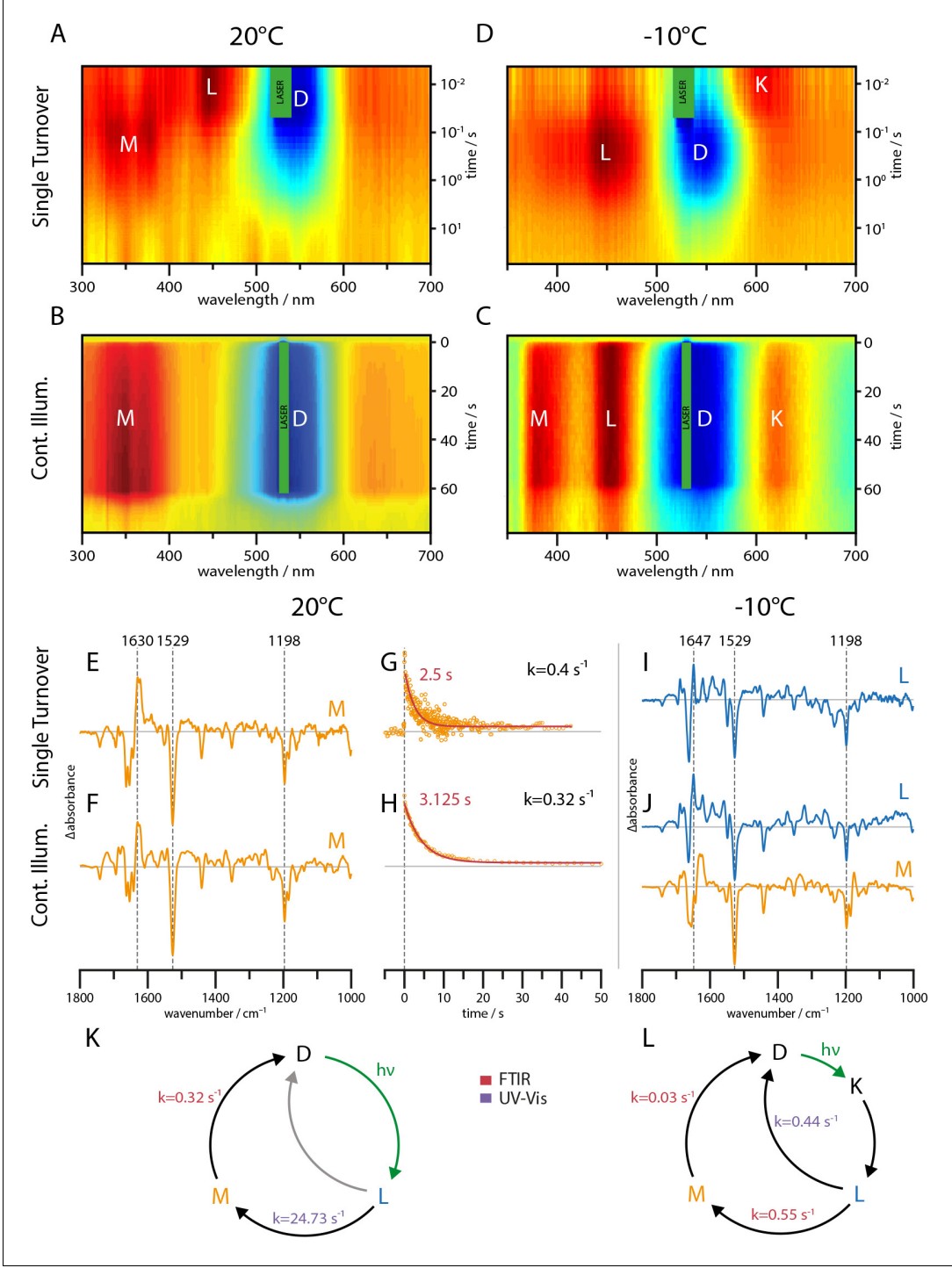

**Figure 1.** UV-Vis (**A–D**) and global-fit analysis of FTIR (**E–J**) difference spectra of the RGC slow mutant C259S for 20°C and −10°C under continuous and single turnover 532 nm illumination. (**A–C**) Contour plot of transient UV-Vis absorption changes. (**E**) Single-turnover spectrum at 20°C corresponding to the M intermediate. (**F**) Decay component after green illumination at 20°C showing features of the M state. (**G**) Kinetic trace of the transient absorption changes after a single green nanosecond laser flash (orange) with corresponding mono-exponential fit (red line). (**H**) Kinetic trace after green illumination (orange) with mono-exponential fit (red line). (**I**) L state spectrum acquired after a single laser flash at −10°C. (**J**) Global-fit analysis of the SVD decay components after continuous green illumination at −10°C showing a L (blue line) and M state (orange line) spectrum. (**K–L**) The implied

*Figure 1 continued on next page*

*Figure 1 continued*
photocycles and corresponding kinetic constants derived via a global fit analysis of FTIR and UV-Vis spectra for 20°C and −10°C respectively.

becomes visible due to the reduced photocycle kinetics, followed by L state formation (*Figure 1D*). However, there is no longer an indication for a later population of the M state, although the protein reverts back to the dark state.

## FTIR spectroscopy – L and M state

The UV-Vis spectroscopic data shows that L and M states can be stabilized under suitable conditions in separate experiments. In parallel to the measurements shown in *Figure 1A*, FTIR spectra of the C259S mutant were recorded with a time resolution of 29 ms and analyzed via rotation and global fit procedure. The resulting FTIR light minus dark spectra and decay kinetics at 20°C are shown in *Figure 1E–H*. The spectral features and decay kinetics at 20°C for single turnover and continuous illumination are similar, implying that the long-lived M intermediate accumulates under continuous illumination. Consequently, the steady state spectrum recorded at 20°C (*Figure 1F*) represents a nearly pure M state spectrum that is virtually identical for single turnover and steady state illumination (*Figure 1E,F*). Due to the similar decay kinetics, we exclude a major branching of the photocycle at room temperature (*Figure 1G,H*). In contrast to the UV-Vis study, we did not obtain FTIR spectra of the L-state at 20°C due to the lower time-resolution. At −10°C, single turnover illumination by short flashes leads to a FTIR difference spectrum that is clearly distinguishable from the spectrum recorded at room temperature. Based on the simultaneously recorded UV-Vis data, we assign this spectrum to the L intermediate. In case of steady state illumination at −10°, the situation is more complex. After illumination stop, a global fit analysis of the decay revealed the presence of two distinct spectroscopic components (*Figure 1J*). The upper spectrum (blue line) shows features of the L-state spectrum, whereas the lower one (orange line) shows great congruence with the M state spectrum obtained at room temperature. As the K state is only weakly populated and expected to rapidly decay, it could not be identified in the FTIR data. These results show the aptitude of two major bands in the amide I region to serve as marker bands for the L and M states (L: 1647(+), M: 1630(+)).

## Shortcut in the photocycle

The global fit analysis of the UV-Vis and FTIR data revealed the kinetic constants depicted in *Figure 1K and L*. While at −10°C, the M state decay is slowed down by only one order of magnitude compared to 20°C, the L state decay is decelerated by a factor of 50. Nevertheless, the M lifetime remains an order of magnitude longer compared to L, which leads, assuming a sequential photoreaction, to an M state accumulation under continuous illumination. Since the M intermediate could not be identified in the single turnover experiment and is less represented than the L intermediate under continuous illumination at −10°C, we propose a L to D relaxation pathway that is populated at lower temperatures but not significantly at room temperature. The presence of two distinct L states, one relaxing back to the dark-adapted state and one leading to M state formation, can be excluded by the identical FTIR L state spectra under continuous and single turnover illumination at −10°C.

## Conformational changes followed by FTIR

*Figure 2* shows the steady state difference spectra of the full-length RGC (aa 1–626) in absence of GTP substrate and of the rhodopsin segment (Rh, aa 1–396) without linker but with a complete N-terminus. All spectra were acquired at 4°C under continuous illumination and scaled upon the chromophore bands at 1220 and 1198 cm$^{-1}$. Experiments on the full-length RGC with GTP substrate did not reveal any differences and are therefore not shown. This apparent lack of enzyme activity is attributable to the dark activity observed in the HPLC activity measurements which were conducted as control (Appendix 6). During equilibration in the spectrometer, the available GTP has already been turned over and is therefore unobservable in the difference spectrum. The RGC wild type (WT) and RGC C259S spectra show great spectral congruence, indicating that despite the altered reaction kinetics, the structural changes remain nearly unaltered. Similarly, the full-length RGC and Rh spectra

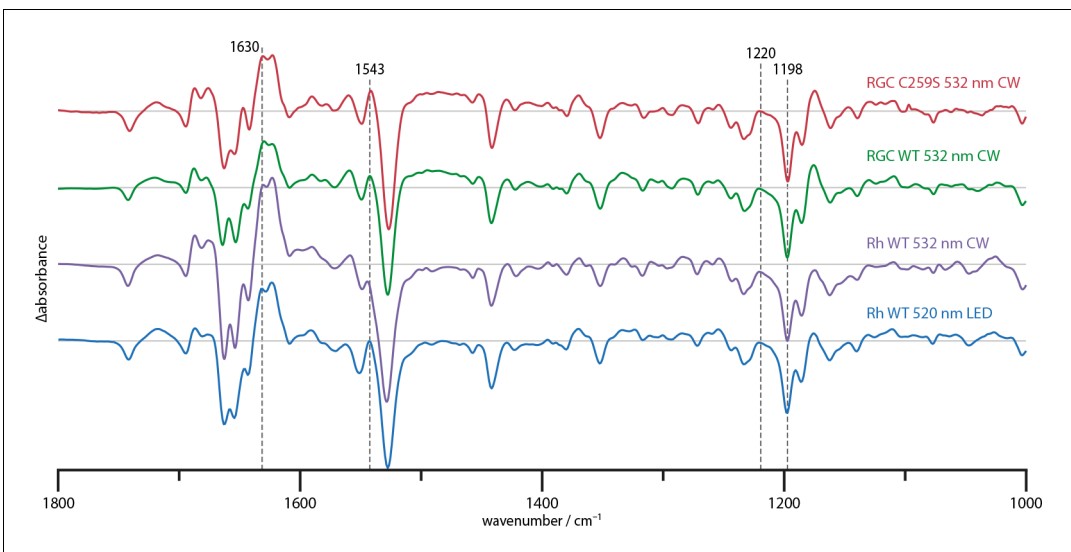

**Figure 2.** Steady state difference spectra of Rh WT illuminated with broad bandwidth 520 nm LEDs and a narrow bandwidth 532 nm CW laser, the full-length RGC WT and the RGC C259S mutant under CW illumination. The spectra are scaled on the 1220 and 1198 cm$^{-1}$ bands in the retinal fingerprint region.

show no obvious qualitative differences and only minor deviations around the amide I (1700–1600 cm$^{-1}$) and II (1590–1510 cm$^{-1}$) regions. Thus, we conclude that the changes in the cyclase domain (GC, aa 443–626) conformation during light-activation are too small to be identified in the spectra as global structural changes.

Previous studies demonstrated that the N-terminus plays a crucial role for the catalytic activity of RGCs, and both C-terminal and N-terminal GC interactions have been proposed (*Scheib et al., 2018*; *Butryn et al., 2020*). Therefore, we analyzed different truncations to elucidate potential structure rearrangements in the linker and N-terminal region during photoactivation. An illustration of the *Ca*RGC structure and truncation sites is given in *Figure 3A*.

*Figure 3B* shows the steady state difference spectra of *Ca*Rh with linker domain and complete N-terminus (RhL, aa 1–442, green line) and a variant with truncation of the N-terminal residues 1–43 (Rh-43, aa 44–396, light green area), without linker domain. The spectra exhibit no particular differences indicating no substantial interaction between the truncated part of the N-terminus and the linker domain or structural changes in the linker domain upon photo-activation. Also, the truncation of these residues seemingly does not lead to changes in the photo-induced behavior of the protein. Our observation is in accordance with the previous finding that this truncation does not impede the catalytic activity of RGC but rather increases it (*Scheib, 2019*). In *Figure 3C*, the same experiment is repeated for the rhodopsin domain truncated at position 140 directly before TM0 (Rh-139, aa 140–396, red line). Compared to Rh-43 (light red area), the overall absorption pattern remains almost unchanged. The presence of the 1630(+) band in both spectra indicates accumulation of the M state at 4°C in both variants. However, smaller structural differences occur in the Rh-139 variant as concluded from the altered amide I and II regions. The intensification of the amide II 1542(+) band in the Rh-139 spectrum causes an apparent shift of the adjacent 1529(-) C=C stretching vibration of the retinal to 1527 cm$^{-1}$. The 1643(-) M-state band in the Rh-43 spectrum has vanished in the Rh-139 variant and instead a band at 1647(+) appears in Rh-139. This data provides evidence for photo-induced structural alterations in the N-terminus or Rh domain due to truncation of the residues 43–139. It indicates an involvement of the cytosolic N-terminus in the photo-induced structural rearrangements of RGC that may support signal propagation from the Rh to the GC domain. This is supported by the complete absence of enzymatic activity in the RGC-139 variant (*Scheib, 2019*).

## Monitoring enzyme activity by FTIR

The RGCs enzymatic functionality has been successfully demonstrated upon coexpression with cGMP-selective CNG-A2 channels in *Xenopus* oocytes and hippocampal neurons (*Scheib et al.,*

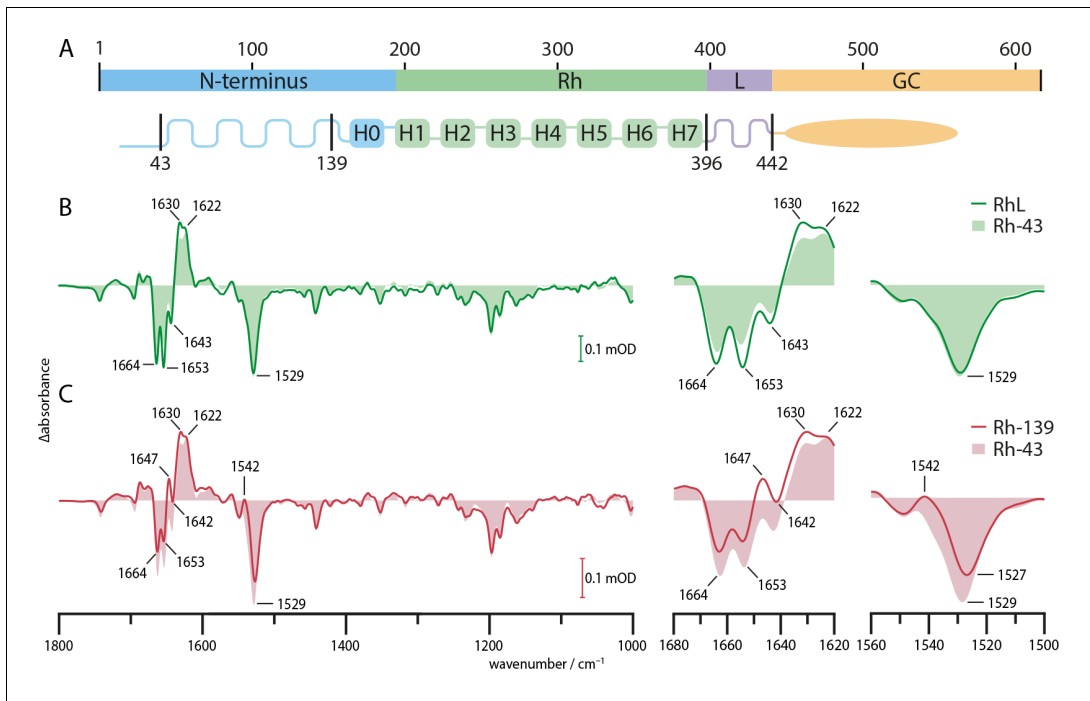

**Figure 3.** FTIR spectra of truncated variants. (**A**) Schematic representation of the predicted *Ca*RGC structure with indicated truncation sites. (**B**) Steady state FTIR difference spectra of RGC with linker domain (RhL, aa 1–442, green line), with slightly truncated N-terminus (Rh-43, aa 44–396, light green area) and (**C**) with removed cytosolic N-terminus (Rh-139, aa 140–396, red line) compared to Rh-43 (light red area). The steady state was determined after the 1630(+) band under continuous wavelength laser illumination at 532 nm. Amide I (1620–1680 cm$^{-1}$) and II (1500–1560 cm$^{-1}$) regions are magnified.

*2015*; *Gao et al., 2015*). To obtain more detailed information about the catalytic mechanism, we applied FTIR transmission spectroscopy in conjunction with caged substrates to address the RGCs reaction mechanism in a direct approach on a molecular level. To verify enzymatic activity, protein of all samples was also tested via an HPLC assay (Appendix 6) which demonstrated on the one hand the necessity to use a photolabile caged GTP substitute which is not turned over by the RGC dark activity and on the other hand, that cGMP is successfully produced by free GC when illuminating the caged NPE-GTP compound with UV-light.

To elucidate the regulatory role of the Rh domain, we compared the substrate turnover of the constitutively active isolated GC enzyme domain (aa 443–626) and the light-activated construct RGC-43 (aa 44–626). Both experiments were initiated by light-induced uncaging of the non-process-able caged compound NPE-GTP into a processable GTP and free nitroacetophenone (NAP). In analogy to the FTIR study on H-*Ras* P21 (*Cepus et al., 1998a*), UV irradiation cleaved the NPE-GTP bond with subsequent initiation of GTP to cGMP conversion by GC or photo-activated RGC. The principal reaction is illustrated in *Figure 4A* and the crystal structure of the GC binding pocket with bound GTP is depicted in B. It has to be noted that the structure was solved in presence of Ca$^{2+}$ instead of Mn$^{2+}$, the latter being mandatory for GC activity. Therefore, the model does not resemble the enzyme catalytic site in its fully active conformation. *Figure 4C* shows the FTIR protocol we have used to monitor the substrate turnover in GC and RGC. For both proteins, NPE-GTP photolysis was initiated by irradiating the sample for 5 s with a pulsed UV laser. In the case of GC, protein and nucleotide changes were monitored for 364 s after photolysis. The RGC required a more complex protocol. After a dark period of 120 s, the NPE-GTP was uncaged and potential GTP processing was observed for 85 s, while the RGCs rhodopsin remains in its dark state, keeping the GC domain inactive. To start GTP to cGMP conversion, RGC was activated with a 532 nm CW laser for 300 s. Finally, the system evolution was recorded for another 364 s in the dark. The NPE-GTP into GTP conversion difference spectrum in presence of Mn$^{2+}$ (*Figure 4D*, light blue area) is dominated by the changes

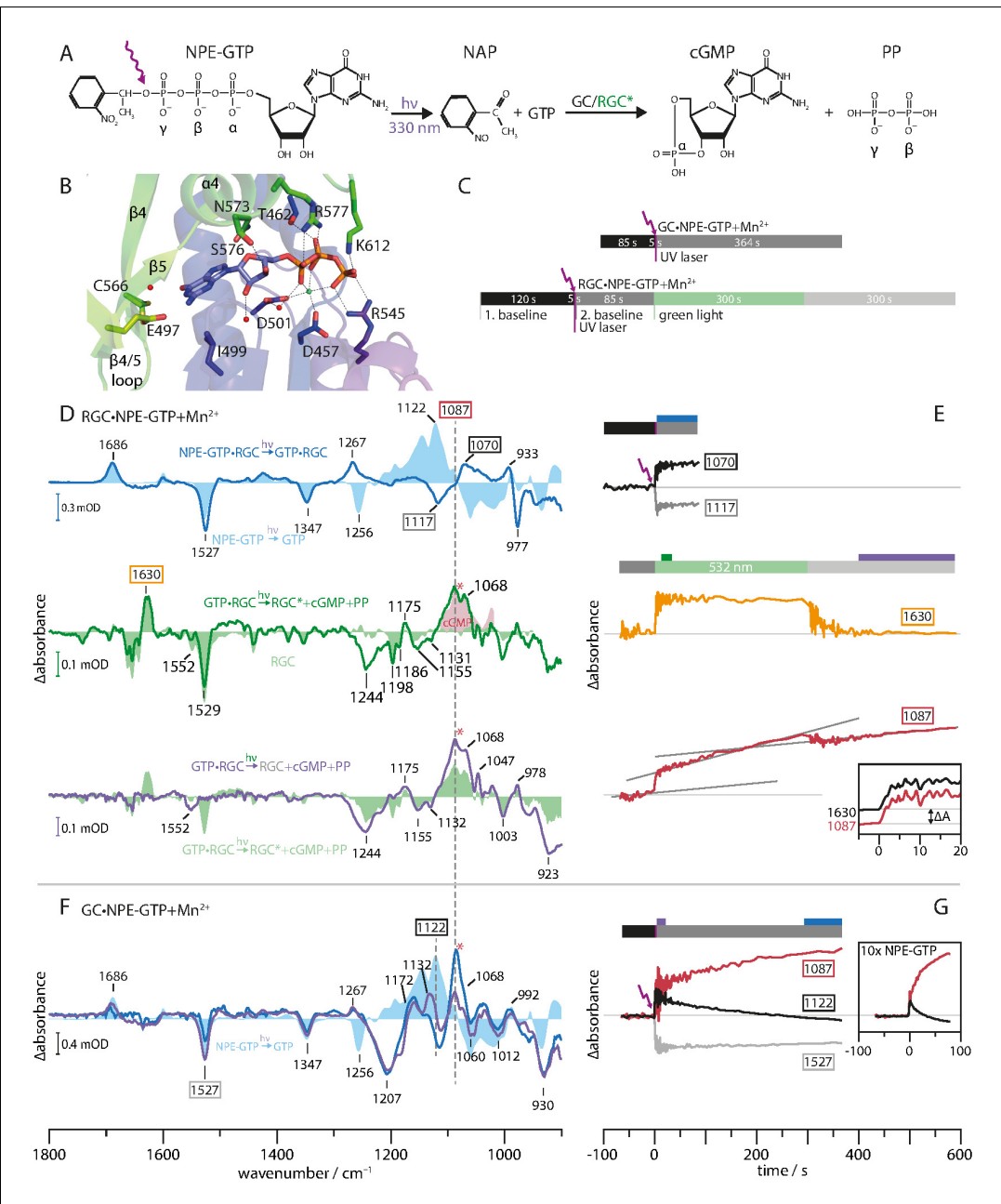

**Figure 4.** Monitoring substrate turnover with FTIR. (**A**) Molecular mechanism of the two-step experiment to monitor GC and RGC enzyme activity. (**B**) *Ca*GC crystal structure with bound GTP (PDB 6SIR). The $Ca^{2+}$ ion is shown as green and water molecules as red spheres. (**C**) FTIR measurement protocols for GC and RGC experiments corresponding to E and G. Colors indicate time intervals over which FTIR spectra in D and F were averaged. (**D**) RGC·NPE-GTP (blue line) and NPE-GTP (light blue area) photolysis FTIR difference spectra. Spectrum averaged from 10 to 20 s (green line) after green illumination start and RGC steady state spectrum (light green area). Clipping of cGMP absolute spectrum (light red area). Spectrum averaged over last 200 s (purple line) compared to green light activation (light green area). (**E**) Kinetic traces of marker bands 1070(+) and 1117(-) (binding), 1630(+) (RGC activation) and 1087(+) (cGMP formation). Gray lines represent sections of constant slope during different illumination conditions. (inset) Normalized, smoothed (eight points) and dark activity corrected kinetic traces of 1087(+) (red line) and 1630(+) (black line) bands at green illumination onset. (**F**) UV-light-induced FTIR difference spectra of GC·NPE-GTP. Averaged spectra over first 20 s (purple line) and last 64 s (blue line) after photolysis. (light blue area) Photolysis spectrum of NPE-GTP. (**G**) Kinetic traces of GC·NPE-GTP photolysis marker bands 1527(-) (NPE cleavage), 1122 and 1087(+) (GTP to cGMP turnover). (inset) Experiment with 10-fold increased NPE-GTP concentration.

of the $NO_2$ vibrations at 1527 and 1347 cm$^{-1}$, a positive band at 1686 cm$^{-1}$ due to the formation of the keto C=O group (*Cepus et al., 1998b*), and a complex difference pattern between 1000 and 1300 cm$^{-1}$, thus indicating rearrangements of the triphosphate cluster. The disappearance of the γ-phosphate $PO_2^-$ vibration due to the NPE-GTP cleavage is indicated by the solitary negative band at 1256 cm$^{-1}$, while the newly formed $PO_3^{2-}$ group is seen by its stretching vibration at 1122 cm$^{-1}$ (*Cepus et al., 1998b*).

In the presence of RGC (*Figure 4D*, solid blue line), the bands originating from the cleavage of the NPE group (1347(-), 1527(-), 1686(+)) are widely conserved after uncaging of NPE-GTP·RGC, while the band pattern in the phosphate region is altered compared to free NPE-GTP uncaging (*Figure 4D*, light blue area). For NPE-GTP·RGC, the phosphate bands show a broader shape and smaller signal intensities when normalized to the NAP band at 1686 cm$^{-1}$. This indicates the reorganization of the GTP phosphate chain to be less extensive in NPE-GTP·RGC and significantly different to free NPE-GTP in solution. Therefore, we conclude that NPE-GTP binds to the RGC moiety prior to UV illumination, thus avoiding major phosphate rearrangements. This conclusion is further supported by the fact that only negligible spectral changes in the carboxylic region >1690 cm$^{-1}$ had arisen, which would be expected for protein carboxylic acids involved in GTP-binding. Obviously, the interaction between the guanine group and E497 (*Figure 4B*) exists already in the dark and remains unchanged upon illumination. The negative band at 1256 cm$^{-1}$ indicating disappearance of the γ-phosphate $PO_2^-$ group due to the NPE-GTP cleavage is not observed, which argues for this vibrational mode to be stabilized by the GC moiety preventing the dipole moment to change and thus erasing its infrared absorption. According to the crystal structure of the GC domain (*Figure 4B*), this could be accomplished by the binding to the active site residues R545 and K612 as well as the metal ion complexation involving the γ-phosphate. The 1117(-) to 1070(+) down shift may be interpreted as a weakening of the P=O vibrations due to slight reorientation of the α- and β-phosphates in the binding pocket after uncaging. This is emphasized by their kinetic traces (*Figure 4E*), indicating a stable conformation which can be attributed to substrate binding rather than turnover. The positive band at 1267 cm$^{-1}$ in the spectrum of the NPE-GTP·RGC complex is assigned to a vibration of the released NAP group, since it also occurs in the photolysis spectrum of the non-hydrolysable GTP analog GPCPP with and without GC complexation (Appendix 2). Here, the $PO_3^{2-}$ mode at 1256(-) band is shifted to lower wavenumbers and thereby reveals the 1267(+) band. Next, we analyzed the molecular processes linked to the RGC activation and to cGMP formation in particular. To this end, the light-induced changes were highlighted by subtracting the uncaging spectrum (*Figure 4D*, blue line) making it the new baseline. The comparison of the spectrum averaged over 10–20 s after illumination onset (green line) with the light-dark difference spectrum from RGC without substrate (light green area) shows great spectral congruence between 1300 and 1750 cm$^{-1}$, demonstrating similar light activation of the Rh domain in both samples. Most strikingly, the 1529(-), 1198(-), and 1186(-) bands indicate retinal isomerization, while the 1630(+) band marks the formation and accumulation of the M state. The main alterations between RGC with and without bound GTP are found in the region of the phosphate vibrational modes below 1300 cm$^{-1}$, revealing initiation of the catalytic process upon rhodopsin activation. This applies to the strong positive 1087 (+) and 1068(+) bands, representing the P=O vibrations arising with the formation of the cGMP. This assignment is based on the great congruence with the cGMP absolute spectrum shaded in light red. *Figure 5A* shows the complete absolute spectra of GTP, cGMP, and PP. As seen in direct comparison, only the 1068 band is unique to the cGMP spectrum, allowing an assignment to the cyclic phosphate bond. Due to its higher intensity, we employ the 1087(+) band as a marker for cGMP production. After illumination onset, the rapid rise of the 1087(+) vibration (*Figure 4E*, red line) reflects the high initial cGMP production, which then levels out to a reduced but constant slope. We explain this observation as a fast GTP to cGMP conversion of the already bound substrate followed by a slow subsequent GTP binding and turnover in our highly concentrated and viscous samples. The instantaneous cGMP production upon formation of the M-state is visualized by almost identical kinetics of the M-state vibration at 1630(+) and the cGMP vibration at 1087(+) during illumination onset, which are shown as inset in *Figure 4E*. The small cGMP dip after green light-off can be explained by the reduced vibrational strength of the cGMP in solution compared to bound cGMP in the GC-binding site. The grey lines, which are fitted to the kinetic sections of approximately constant slope, reveal a substantial dark activity before and after RGC light-activation. The cleavage of the

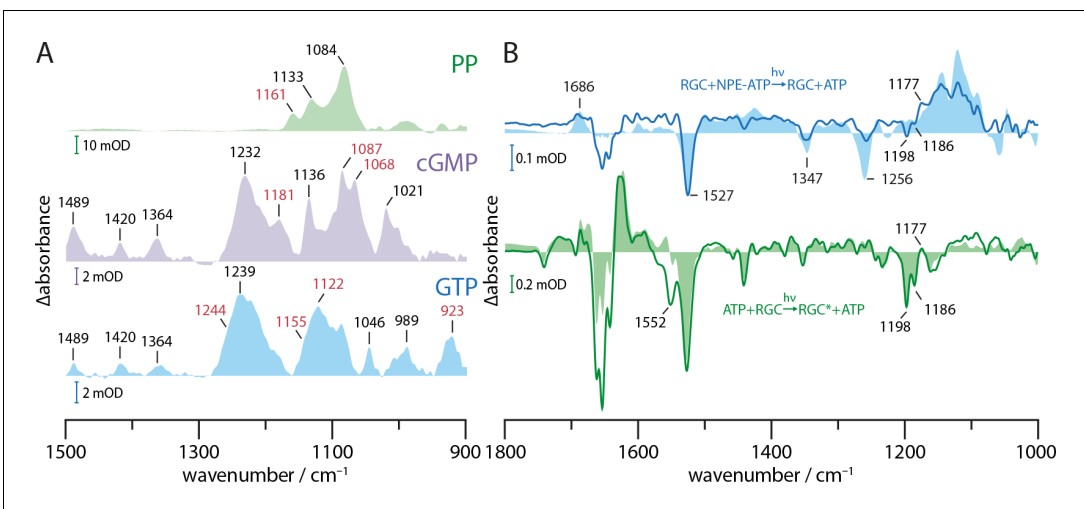

**Figure 5.** Substrate turnover supplementary studies. (**A**) Absorption spectra of PP, cGMP, and GTP in presence of $Mn^{2+}$ in a substrate to ion ratio of 1:2. The spectra were baseline corrected and smoothed using 25 spectral points. Bands discussed in context with GTP to cGMP and PP turnover by GC and RGC are shown in red. (**B**) FTIR difference spectra of catalytic interaction of RGC with NPE-ATP in the presence of $Mn^{2+}$ measured with the same protocol as shown in *Figure 4C*. NPE-ATP+RGC to ATP+RGC photolysis spectrum (blue line) was derived via global fit analysis and has been smoothed using 18 spectral points due to the poor signal-to-noise-ratio. For comparison, an NPE-ATP+$Mn^{2+}$ photolysis spectrum (light blue area) is underlaid. Using the spectra measured for 85 s after UV-light-induced photolysis as baseline the RGC photo-activation difference spectrum averaged over 20–300 s during green illumination with a 532 nm CW laser is shown (green line). A Rh steady state difference spectrum is underlaid for comparison (light green area).

phosphodiester bond between α- and β-phosphate associated with GTP to cGMP conversion is reflected by the negative 1244(-) and 1155(-) bands (*Figure 4D*).

After light-off, rhodopsin inactivation is indicated by the disappearance of the prominent Rh bands above 1400 $cm^{-1}$ (*Figure 4D*, violet line) and 1630 M-state marker band kinetic shown in *Figure 4E* (orange line). In parallel, the cGMP production is initially diminished but then further increases with reduced speed during the 300 s in darkness. Moreover, the negative 923(-) band gains intensity upon cGMP release. The absence of absorption bands in the cGMP and PP absolute spectra allows an assignment to the GTP P-O-P vibration (see *Figure 5A*), leading to a reduced absorbance over time whilst the GTP is turned over. Finally, we performed uncaging experiments with the cyclase·NPE-GTP complex (GC·NPE-GTP). The FTIR difference spectra of GC-bound caged-GTP during the early (0–20 s, purple line) and a later phase of GTP conversion (300–364 s, blue line) after NPE-GTP uncaging are shown in *Figure 4F*. Besides the expected disappearance of the NPE-GTP vibrations at 1347(-) and 1527(-), we see phosphate modifications, which on the one hand are clearly different from free GTP (light blue area) and on the other hand differ from the conversion of GTP into cGMP in light-activated RGC (*Figure 4D*, green line). Differences found in comparison to the photolysis spectra of free GTP (light blue area) are seen in the phosphate region below 1300 $cm^{-1}$. Especially the negative band at 1256 $cm^{-1}$ associated with the γ-phosphate $PO_2^-$ stretching vibration appears to be displaced at 1207 $cm^{-1}$, indicating that also in this sample the NPE-GTP was almost completely bound to the protein prior to cleavage. The $PO_3^{2-}$ stretching vibration, formed upon photolysis, which was found at 1122 $cm^{-1}$ for free GTP (*Cepus et al., 1998b*), appears upshifted to 1132 $cm^{-1}$. Similar shifts were observed for *Ras* bound NPE-GTP upon uncaging (*Cepus et al., 1998a*). The bands at 1087 and 1068 indicate formation of cGMP, as observed in the GTP·RGC spectrum, and were found consequently with increasing intensities when the time progresses after uncaging. The different shape of the cGMP bands in GC compared to RGC could be caused by altered band intensities due to different buffers. In GC, the dominant 1084 PP band shown in *Figure 5A* seems intensified, leading to minor downshift of the 1087 band and apparent fusion of the 1087 and 1068 bands. The positive band at 1172 $cm^{-1}$ gaining intensity after photolysis most likely originates from the cyclic GMP and pyrophosphate formation (PP: 1161 $cm^{-1}$, cGMP:

1181 cm$^{-1}$, *Figure 5A*) and could correspond to the 1175(+) band in RGC. The 1087 and 1122 band kinetics depicted in *Figure 4G* show the instantaneous GTP uncaging as a step. This is then followed by cGMP production, as the 1087 band rises and simultaneous GTP disappearance as is indicated by the decline of the 1122 band. The enzymatic process was found to already be saturated during the measurement. An analog experiment with 10 times higher NPE-GTP concentration (*Figure 4G* inset) shows that the reaction kinetic and its saturation are accelerated, suggesting feedback inhibition by the reaction product cGMP. The positive 1267(+) band is a shared feature of both enzymes, inactive RGC and active GC (*Figure 4D,F*). However, it was not found to be visible in the NPE-GTP spectrum due to superposition with the 1256(-) band in free NPE-GTP (light blue area). Since the same band originates in the photolysis spectrum of the GTP analog NPE-GPCPP (Appendix 2), we assigned this band to the UV-light-induced release of the NAP-group. Differences in the phosphate region in the GTP·RGC complex between 1300 and 1000 cm$^{-1}$ compared to those in the spectrum of the GTP·GC complex reflect differences of GTP binding to an inactive (*Figure 4D*, blue line) and an active cyclase domain (*Figure 4F*, blue line). This may also hold for the 1207(-) band observed only after photolysis of NPE-GTP in presence of GC, indicating a preserved γ-PO$_2^-$ mode only in the inactive NPE-GTP·RGC complex. *Figure 5B* shows the FTIR difference spectra of the RGC upon UV-uncaging of NPE-ATP in presence of Mn$^{2+}$. The protocol was analogous to that for NPE-GTP uncaging (*Figure 4C*). The photolysis spectrum (blue line) shows an overall congruence with a photolysis spectrum acquired for uncaging of free NPE-ATP in solution despite the reduced signal quality. This observation hints towards a caged ATP, which is, unlike caged GTP, unable to form tight bonding to the dark-adapted RGC active site. Accordingly, the low signal to noise ratio is caused by the non-bound substrate being mostly removed during concentration in the centrifugal filter (Materials and methods). A different binding mode of NPE-ATP to RGC, that in contrast to GTP is unable to interact tightly with E497 and C566, is further indicated as no shift of the negative 1256 band is observed. Additionally, the NPE-group impairs the correct orchestration of the phosphate chain effectively inhibiting substrate retention in the active site.

## Discussion

### Characterization of photocycle intermediates

In the experimental and supplementary sections, we described a versatile setup for simultaneous UV-Vis and FTIR recording of laser-induced chromophore changes and structural developments within the enzymerhodopsin RGC from *Catenaria anguillulae*. Utilizing the slow cycling C259S mutant, characterized by its decelerated photocycle kinetic, we obtained high resolution FTIR spectra of the L and M photocycle intermediates of *Ca*RGC which allowed us to assign characteristic FTIR marker bands. In accordance with our findings, flash photolysis experiments on *Ca*Rh WT showed a sequential photoreaction at room temperature (*Scheib et al., 2018*). The time constants were reported to be $\tau_L$ = 30 ms and $\tau_M$ = 570 ms resulting in a 5.5 times retarded M state decay for the C259S mutant ($\tau_M$ = 3.13 s) while the L state kinetic remains nearly unaffected ($\tau_L$ = 40 ms). Interestingly, the M state spectra of the mutant and the WT (*Figure 2*) do not exhibit any obvious differences which might hint towards the causal root of the deceleration. A detailed spectroscopic analysis of this phenomenon is currently underway. Despite the clustering of common features, no clear evidence for the population of parallel photocycles starting from 13-*trans*, 15-*anti* or 13-*cis*, 15-*syn* retinal was found for *Ca*RGC as reported for ChR2 (*Kuhne et al., 2019*). The slowly increased accumulation of the M state during continuous illumination however (see *Figure 1B*), could be an indication of a more complex light adaptation.

The L-state FTIR difference spectrum (see Appendix 3) is dominated by a 1665 (-)/1647(+) cm$^{-1}$ double band which suggests hydration of a transmembrane α-helix within the Rh moiety due to water influx, as proposed for Channelrhodopsin (*Lórenz-Fonfría et al., 2015*). In many microbial rhodopsins, deprotonation of the RSB during the formation of the M-state is accompanied by major conformational changes leading to functional events such as changes of the surface accessibility or channel opening/closing. In *Ca*RGC, the FTIR difference spectra of the M-intermediate show prominent bands at 1622, 1630 and 1677, 1688 cm$^{-1}$ indicating conformational changes that may be interpreted as formation of new β-sheet structures in the rhodopsin domain. Besides confirming a classical sequential photocycle involving an M-state with deprotonated RSB, our experiments

showed that, at least at low temperatures, the recovery of the RGC dark state also proceeds directly from L via a second relaxation pathway (short cut). Based on our FTIR data, both L-state decay paths, either directly back to the dark state or via the M-state, have comparable rate constants at −10°C, indicating a metastable L-state. In contrast, the putative β-sheet conformation in the M-state is thermodynamically more stable and thus better suited to trigger enzyme activation. While the M-state was proposed as the catalytically active state of RGC (*Scheib et al., 2018*), our FTIR study confirmed this assumption by revealing a direct correlation of the M-state and cGMP formation indicated by the kinetics of the respective marker bands at 1630 cm$^{-1}$ (M) and 1087 cm$^{-1}$ (cGMP). Remarkably, FTIR difference spectroscopy assigns the vast majority of the structural rearrangements occurring after photoactivation to the rhodopsin domain, while the contributions of linker and GC domain to the spectral features are marginal.

Clearly distinct activation mechanisms have evolved for different type III cyclases. For example, in soluble NO sensing GC (sGC), NO binding induces major conformational changes (*Kang et al., 2019*), whereas the structural alterations in a soluble light-activated adenylyl cyclase from cyanobacteria upon activation are minor and therefore difficult to track (*Ohki et al., 2017*). Previous studies on type III cyclases report the GC dimer activity to be regulated by a small angular repositioning of the two subunits towards one another that leaves their internal structure and overall peptide environment nearly undisturbed (*Butryn et al., 2020*; *Krissinel and Henrick, 2007*; *Steegborn, 2014*). Accordingly, a slight rotation of TM7 could propagate through the linker and affect the GC domain in analogy to the sensory rhodopsin II signaling pathway (*Gordeliy et al., 2002*). This would leave the linker environment nearly unaltered and therefore would not result in vibrational changes identifiable in the FTIR difference spectrum. Our data shows, however, light-induced structural changes in the M state that involve the N-terminal residues 44–139. It indicates a stronger water exposition of secondary structure elements, resulting in reduced amide modes. These could be located in the N-terminus, caused by a change in helix −1 or an unraveling of the random coil structure, or by altered structure rearrangements in the Rh domain due to the truncation. Since removal of the cytosolic N-terminus (RGC-139) results in the complete loss of enzymatic function (*Scheib, 2019*), the linker contact has to hold the GC domain in an catalytic inactive position. Consequently, the N-terminus mediates the activation by either releasing the inhibitory linker contact or by directly altering the GC conformation. Since it has been shown that light-activated RGC is more active than free GC alone we presume the latter (*Mukherjee et al., 2019*). As our results suggest, the N-terminal rearrangement in the M state introduces small but crucial modifications of the active site as a requirement for cyclase activation (*Scheib et al., 2015*; *Kumar et al., 2017*; *Alben et al., 1974*).

The functional involvement of the C-terminal linker domain however, could not be observed in the FTIR experiments. Nonetheless, a signal propagation via the linker domain and the shorter handle helices at the GC surface cannot be excluded. Structural alterations in this segment could preserve the peptide environment, thus rendering a possible movement unobservable by the method. Slight rotations or a scissor like movement in a constantly hydrated environment would not precipitate strong amide mode changes. Smaller amide features however, could be masked behind more dominant band patterns of the Rh segment. Since highly conserved linker constructs are involved in GC and AC regulation (*Ma et al., 2010*; *Vercellino et al., 2017*; *Ziegler et al., 2017*; *Qi et al., 2019*; *Khannpnavar et al., 2020*; *Seth et al., 2020*), a similar mechanism cannot be disregarded to play a role in RGCs as well. The *Ca*RGC transducer element is predicted to form a structural motif consisting of an elongated TM and a short handle helix which is most similar to the S-helical domains found for example in Gsα. Therefore, signal propagation presumably requires a concerted reorientation of both, linker domain and N-terminus.

## Enzyme activity

To achieve a direct correlation of vibrational changes within the protein and the substrate processing during enzyme activation, we tracked the GTP to cGMP turnover of the isolated GC domain and the photoactivated RGC by FTIR spectroscopy. They disclosed that NPE-GTP already binds to the dark-adapted RGC catalytic site, ruling out that photo-activation regulates the accessibility of the binding site as discussed for other type III cyclases (e.g. sGC) (*Kang et al., 2019*). Analogous experiments with NPE-ATP show lower binding of caged ATP to the protein compared to GTP, suggesting that base interactions are involved in substrate binding. This is remarkable, because so far, base-binding has been assigned to occur rather late during catalytic turnover, whereas substrate binding was

considered to be dominated by the protein-metal-phosphate interaction (*Steegborn, 2014*). The fact that RGC is able to bind GTP in the dark state without the need to open and close the active site, is consistent with our observation that little structural rearrangement of GC occurs upon photo-activation. The main regulation of catalytic function seems to take place in the active site. We found that NPE-GTP also binds to the isolated GC domain. However, the phosphate bands between GC and dark-adapted RGC are altered as a result of differing active site conformations in the dark-adapted, hence inactive RGC and the inherently active GC. In the isolated GC, well-separated phosphate bands were observed in contrast to the broader bands in the inactive RGC, thus indicating that the presence of the dark-adapted Rh domain prevents vibrational uncoupling of the phosphate vibrations. This uncoupling has been observed for *Ras* bound GTP as a prerequisite of phosphate chain cleavage (*Allin et al., 2001*). The experiments also revealed a substantial RGC dark activity in detergent which does not exist for RGC in host membranes and necessitates the use of caged GTP compounds. This may be caused by the residual presence of free GC due to unspecific fragmentation during protein preparation or the high protein concentration in detergent.

The NPE-caging group located at the γ-phosphate modifies the binding site but does not prevent its occupation. Instead, it impedes correct orchestration of the phosphate chain, effectively disabling the formation of the transition state leading to the cleavage of the α-β phosphate bond. GTP preincubation was found to be an experimental requirement since substrate diffusion into the binding site is insufficient in the highly concentrated detergent samples necessary for FTIR transmission spectroscopy. As expected, no substrate turnover was observed in experiments with NPE-ATP, indicating that the purine-base discriminating interaction assigned to the residues C566 and E497 are indispensable for catalysis. It has to be noted that the photolysis spectrum of caged ATP shows signs of a weak rhodopsin activation by the 330 nm laser indicated by the 1198(-), 1186(-), and 1177(+) retinal bands. However, the cross-interaction can be considered low when taking into account the low overall absorption of the spectrum and the absence of the dominant 1630/1622(+) amide I features. Also, the 1552(-) band, previously observed in the RGC-bound caged-GTP experiment, similarly emerges with caged ATP and excludes its connection to substrate turnover. It may be caused by sample heating due to the UV-light irradiation or long-term green light exposure.

## Activation mechanism

Now the question arises about which structural rearrangements in the GTP-binding pocket will lead to cleavage of the α-β phosphate bond and formation of cGMP. As deduced from various functional and structural studies, two metal ions, A and B (*Figure 6*), are involved in the catalysis, believed to be $Mg^{2+}$ under physiological conditions in most type III cyclases (*Steegborn, 2014*; *Tesmer et al., 1999*). While ion B is permanently present and mediates binding of the phosphate moiety of the substrate, ion A resides only temporary and may tune the electrostatic environment of the ribose-3'OH for the nucleophilic attack of the α-phosphate. Notably, both monomers are involved in GTP/ATP binding with the purine-base and the phosphates interacting with different monomers; thus, slight changes of the dimeric arrangement during protein closure may trigger the conformation of the substrate to allow cyclization. Most type III cyclases have remarkable substrate specificity, catalyzing either ATP or GTP, rendering precise nucleotide base orientation crucial. Therefore, the interaction of the protein to the purine holds the potential to regulate cyclase activity. Accordingly, an activation mechanism involving structural reorientation of the β4/5 loop immediately preceding the nucleotide base-contacting β5 region of the cyclase domain has been proposed. This mechanism derived originally from the analysis of various crystal structures of a soluble light-activated adenylyl cyclase from the bacterium *Beggiatoa sp.* (*Lindner et al., 2017*). Different conformations of this flexible loop have also been reported for the crystal structure of a *Ca*GC mutant (*Scheib et al., 2018*), providing some evidence that this region may also be involved in enzyme activation in *Ca*RGC. The same study reported a certain amount of flexibility in the positioning of the residue R577 on helix α4 after β5 (*Appendix 5—figure 1B*) which might indicate a regulatory role in conjunction with the proposed purine-base interaction. The positive charge of R577 is discussed to compensate the temporary additional negative charge at the pentavalent α-phosphate during cyclization (*Steegborn, 2014*). The potential impact of positive charged residues on hydrolysis of GTP has been described in detail for the GTPase reaction of H-*Ras* P21, which produces GDP and $P_i$ (*Cepus et al., 1998b*; *Allin and Gerwert, 2001*). Binding of the GTPase-activating protein (GAP) places a single

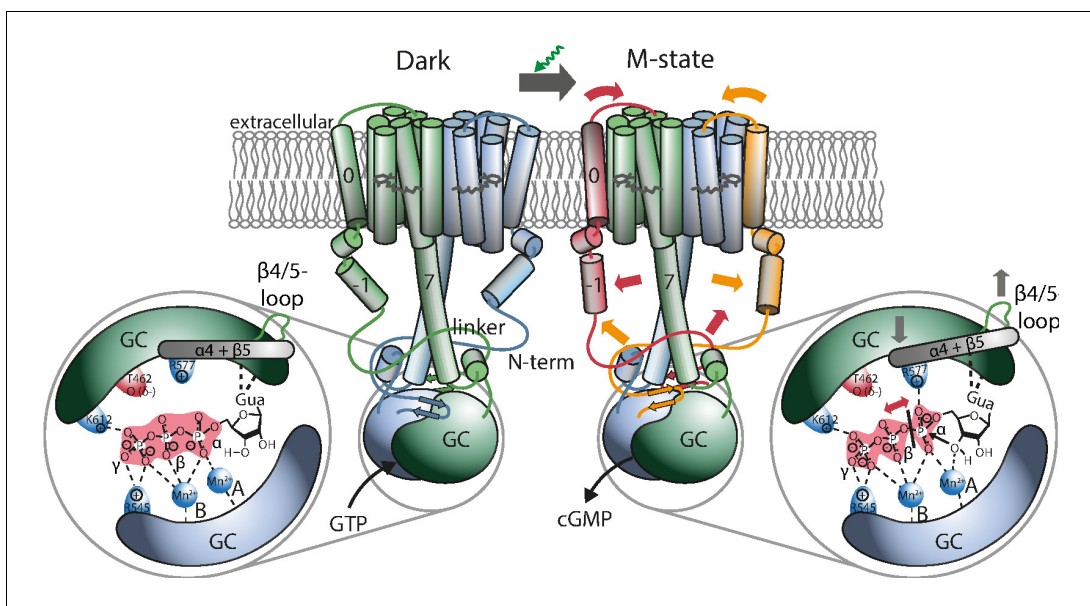

**Figure 6.** Proposed mechanism leading to light-induced activation of the GC domain. GTP binds to the dark-adapted RGC. After photon absorption, the Rh domain converts into the M state causing the N-terminus to interact with the linker or be retracted from the β4/5-loop at the GC domain, which then assumes an active position. The positive charged R577, located at helix α4 after β-sheet 5, interacts with the α-phosphate, making it more vulnerable to the nucleophilic attack of the ribose-3'-OH group leading to subsequent cleavage of the α-β phosphate bond.

arginine residue close to the β-phosphate group thereby accelerating the GTP hydrolysis rate by five orders of magnitude *Allin et al., 2001*; *Rudack et al., 2012*.

To obtain a structural reference for a potential signal propagation from Rh to GC domain, we constructed two homology models (HMs) of the RGC-43 system (Appendix 4-5). One was created with the help of Swiss-Modell *Waterhouse et al., 2018* and Robetta *Song et al., 2013* on the basis of the rhodopsin phosphodiesterase crystal structure from *Salpingoeca rosetta* (*Sr*RPDE, PDB-IDs: 7CJ3, 7D7Q) *Ikuta et al., 2020* for the Rh domain which was linked to the *Ca*GC crystal structure (PDB-ID: 6SIR) *Butryn et al., 2020*. The second structure was derived using the recently published AlphaFold 2 advanced notebook on Google Colab *Mirdita et al., 2021*. Although the AlphaFold 2 model confirmed most predictions about the general structure of the linker domain it exhibits several deviations from the previous model. This includes mainly the placement of helix 0 which is no longer found at the dimeric interface but adjacent to TM2 and 3. This is accompanied by an altered positioning of helix −1 which is now nearly parallel to the linker helix. However, while AlphaFold 2 reports high levels of confidence for the majority of the structure, the confidence for the prediction of helix −1 is reduced and approaches zero for the subsequent disordered part of the N-terminus which leaves its conformation unclear. The Rh domain shares great resemblance to the *Sr*RPDE crystal structure of the TM domain and therefore seems more plausible than the previous model. Also, the AlphaFold 2 prediction shows the linker and helix 8 in much closer proximity to the β4/5-loop which is thereby displaced towards the GC center compared to the crystal structure. This suggests a regulatory interaction. On the other hand, the β4/5-loop remains accessible to a putative interaction with the N-terminal β-sheets. *Figure 6* depicts a schematic protein structure based on both HMs, illustrating a putative mechanism involving a N-terminal interaction with the linker or the β4/5 loop and the resulting changes in the active site: As demonstrated, GTP binds to the active site in the RGC dark state mainly stabilized by the metal ion complex involving D501, D457 as well as R545 and K612. Upon green illumination, RGC reaches the M-state which may result in a tilting motion of TM0. This conformational change propagates towards the GC domain via helix −1 and either changes the linker conformation or leads to the putative interaction of an N-terminal β-sheet (predicted for aa 68–72 and 90–93) and the β4/5-loop of the cyclase domain. The nature of this interaction remains unclear but since no significant spectral influence could be assigned to the GC domain,

the overall changes must be small. Since an involvement of the C-terminal linker helices in the photo-activation could not be observed, potential alterations in this segment are not highlighted in the model. Nevertheless, a concerted action of linker and N-terminus appears possible. The altered contact between the N-terminus and the β4/5-loop would result in a catalytic competent active-site arrangement, which facilitates the correct base-binding and further stabilizes the transition state. The latter likely involves an enhanced interaction of the highly conserved positively charged R577 on α4 with the α-phosphate.

A combined study including FTIR as well as functional and structural data may help to answer further details of light-activation in RGC. Only recently, fungal RGCs have been described that function as obligate heterodimers, which utilizes only one of the rhodopsin domains for direct light-activation *Broser et al., 2020*. This light-activated rhodopsin-cyclase subunit shares up to ~45% sequence identity with *Ca*RGC but is completely inactive as homodimer. Notably, the N-terminus of this rhodopsin is ~80 aa shorter compared to *Ca*RGC and particularly the region aa 43–139 of *Ca*RGC is barely conserved, providing further evidence for a functional role of this region in homodimeric *Ca*RGC.

## Materials and methods

### Experimental setup

While UV-Vis spectroscopy keeps track of the chromophore evolution, IR spectroscopy may provide more detailed information on changes about the protein backbone and individual amino acid residues. Time-resolved UV-Vis and IR spectroscopy on photosensitive proteins require a specific and reproducible experimental environment. This entails mainly full control over temperature and illumination during the measurement to obtain information about vibrational changes that are complementary to the information about electronic alterations in the chromophore. To meet these requirements, a customized experimental setup was developed, which allows us to conduct UV-Vis and IR measurements simultaneously on the same sample. The setup is described in the following paragraph. *Figure 1* of Appendix 1 depicts a schematic view of the setup with its main components and customizations. The base forms the Vertex 80 v FTIR spectrometer (Bruker Optics, Karlsruhe, Germany), which is capable of recording IR spectra in rapid-scan mode with a sampling rate of up to 320 kHz. The spectrometer is equipped with a $N_2$-cooled MCT detector (Kolmar Technologies, Newburyport, MA, USA) for the IR range. Additionally, an Ocean FX UV-Vis array detector (Ocean Optics, Largo, FL, USA) has been installed into the setup. It facilitates data recording in the spectral range from 200 nm to 1100 nm with integration times ranging from 10 μs to 10 s. The UV-Vis beam line has been designed to be parallel to the IR beam path to exclude dispersive effects that are due to refraction caused by the oblique incident light at the sample windows. The interferometer compartment contains a small bandwidth laser to determine the mirror position via a photodiode (16) (Connes-Advantage). The laser is then decoupled via a small prism in the IR beam path. The shadow cast by this prism is now used to couple in and out the UV-Vis light via additional prisms (3) in the beam path. The Deuterium-Tungsten Halogen light source (Dh-2000-BAL, Ocean Optics) is coupled into the spectrometer via an optical fiber. The light is then collimated (1) and focused via a telescope optic onto the prism (3) to ascertain maximal throughput at the sample (5) and minimize the loss of light when coupled out at the final prism before the UV-Vis detector. A collimator at the detector (8) maximizes the yield through the 100 μm entrance slit. Due to the high absorbance of water vapor in the observed spectral range, the entire spectrometer is kept evacuated. The sample holder is mounted on an aluminum sample rod for better accessibility. A circulation thermostat can be connected to it from outside to control the sample temperature by circulating coolant through the rod. The whole assembly is fixed on a stainless-steel bellow to allow sample position adjustment from outside the vacuum chamber. To allow for various illumination conditions, a ring of laser diodes and LEDs can be fixed onto the sample holder and controlled via the OPUS 7.5 (Bruker Optics) measurement software. To allow for time-resolved measurements under single turnover conditions, the sample compartment is accessible to a 10 Hz Nd:YAG Powerlite laser as a pump source for a Horizon II optical parametric oscillator (Continuum, San Jose, CA) from outside via a quartz window (7). The whole ensemble is orchestrated via custom designed software (C++) and electronics allowing for precise synchronization of laser pulses, continuous light sources, as well as IR and UV-Vis data

recording. UV-Vis data was acquired via a custom program written in C# based on the Streamer test software provided by Ocean Optics.

## Molecular biology

Mouse codon-adapted DNA encoding rhodopsin guanylyl cyclase from *Catenaria anguillulae* (*Ca*) (gb: MF939579) 1–626 was ordered from GenScript, USA. Additionally, the sequences encoding N-terminal 43 amino acids were truncated during PCR amplification to create the *Ca*RGC-43. The C259S and other mutations were generated via Quickchange PCR reactions (Agilent) on the *Ca*RGC-43 construct. For expression and purification, wild type or mutated *Ca*RGCs were cloned in pPICZ C yeast expression vector (Thermo Fisher Scientific) under AOX1 promoter, 5′ upstream of a 6×His-tag. For electrophysiological measurements in *Xenopus* oocytes, *Ca*RGC wild type/mutants were sub-cloned via BamHI and HindIII into pGEM (Promega) and cRNA that was produced by reverse transcription (Promega). For expression and purification of the soluble wildtype/mutated cyclase domains (residue 443–626), cyclases were cloned via NdeI and XhoI into pET21a+ plasmid (Novagen), 5′ upstream of the 6×His-tag.

## Protein expression, solubilization, and purification

The *Ca*RGC wild-type/tr-43/mutants were heterologously expressed (pPICZ C) in *Pichia pastoris* cells using the EasySelect Pichia Expression system (Thermo Fisher Scientific) according to the manufacturer's protocol. *Pichia pastoris* clones with the highest *Ca*RGC expression were selected after screening multiple clones by Immunoblotting. Selected clones were then grown in BMGY growth media followed by BMMY induction media supplemented with 5 μM all-*trans*-retinal. Twenty-four hr after Methanol induction, the Pichia cells were harvested, washed with buffer HBS (50 mM HEPES pH 8.0, 100 mM NaCl, complete Protease Inhibitor) and stored at −80°C. All further purification steps were performed at 4°C. Solubilization of the protein was achieved by adding LMNG (2,2-didecylpropane-1,3-bis-β-D-maltopyranoside) and cholesterol hemi-succinate (CHS) to the final concentrations of 1% (w/v) and 0.1% (w/v), respectively, with a protein-detergent ratio of 1:10. After the binding of protein to Ni-NTA resin (5 ml of HIS trap crude column, GE Healthcare) and washing the column with 10 column volumes of HBS buffer with 50 mM imidazole and 0.001%/0.0001% LMNG/CHS, protein was eluted with HBS buffer, 500 mM imidazole and 0.001%/0.0001% LMNG/CHS. Eluted protein was desalted (Hiprep 26/10 desalting column, GE Healthcare), pooled, and loaded on a size exclusion column (HiLoad 16/600 Superdex 200 pg (GE Healthcare)). Purified protein was concentrated with an Amicon Ultra 100 kDa (Millipore) to an optical density of 1 at 540 nm. Concentration of purified protein was determined by absorption at 540 nm, ε = 45,000 $M^{-1}cm^{-1}$. His-tagged truncated cyclase domains (*Ca*GC, protein residues 443-626) were expressed in *E. coli* C41 (DE3, Lucigen) cells at 37°C in 4 × 800 ml culture volume of Luria-Bertani broth, containing 100 μg ml$^{-1}$ ampicillin. At OD600 = 0.5, cells were cooled to 18°C for 1 hr and expression was induced overnight with 1 mM IPTG at 18°C. Cells were harvested and lysed (three rounds of French press). Cell debris was removed by two consecutive centrifugation steps: (1) 10 min at 16,000×g, 4°C and (2) 1 hr at 40,000×g, 4°C. The supernatant was then loaded on a Ni-NTA column (5 × 5 ml HisTrap HP, GE Healthcare). Following a washing step with 20 column volumes of 20 and 50 mM imidazole, the protein was eluted with 500 mM imidazole and buffer-exchanged by a Hiprep 26/10 desalting column (GE Healthcare) that was equilibrated to 20 mM Tris/HCl, 50 mM NaCl, pH 8.0. Fractions of interest were pooled with a 50 kDa Amicon Ultra Filter (Millipore) and put through a size-exclusion column chromatography process (HiPrep 16/60 Sephacryl S-100 HR (GE Healthcare)). After elution with 20 mM Tris/HCl, 50 mM NaCl, pH 8.0, fractions containing the monomeric protein were pooled and concentrated with a 10 kDa Amicon Ultra Filter (Millipore). The concentration was measured (Nano Drop) to have the cyclase molecular weight of 21.5 kDa. (*Ca*GC extinction coefficient: 31,000 $M^{-1}cm^{-1}$ at 280 nm).

## HPLC assay for activity determination

Enzyme activity was tested for all measurements involving substrate turnover, including control measurements. Activity assays at room temperature were carried out in 50 mM HEPES, 100 mM NaCl, pH 8 (final volume 100 μl) with 1 mM of GTP and 2 mM $Mn^{2+}$. One μM full-length RGC protein was either illuminated with green light (522 nm, 0.010 mW mm$^{-2}$, Adafruit NeoPixel NeoMatrix 8 ×

8–64 RGB) during the incubation time or kept in darkness. Enzymatic activity was stopped by flash freezing in liquid nitrogen and an addition of 200 μl 0.1 N HCl. After centrifugation (90 s, 12,000×g, RT), the supernatant was filtered through a 0.2 μm chromafil filter (Macherey-Nagel) and 25 μl was applied on a C18 Reversed Phase High Pressure Liquid Chromatography (HPLC) column (SUPELCO-SIL LC-18-T, 3 μm particle Size, 15 cm × 4.6 mm, Sigma Aldrich), which was equilibrated to 100 mM $K_2HPO^4/KH_2PO^4$, 4 mM tetrabutylammonium iodide, pH 5.9, 10% methanol at a flow rate of 1.2 ml $min^{-1}$. Analyte elution was recorded via an absorbance at 260 nm (retention time for cGMP ~7 min). cNMP was quantified by peak analysis in Origin 8.5.5 (Originlab) and compared to peak values of cNMP standards (Sigma Aldrich) of known concentration or directly compared with another reaction mixture.

For the activity assays of free GC, 1 μM protein was mixed with 1 mM GTP or NPE-GTP and 1 mM $Mn^{2+}$ in a final volume of 100 μl. The incubation time was 10 min. Enzymatic activity was stopped by flash freezing in liquid nitrogen and addition of 200 μl 0.1 N HCl. After centrifugation (90 s, 12,000×g, RT), the supernatant was filtered through a 0.2 μm chromafil filter (Macherey-Nagel) and 25 μl was applied on a C18 Reversed Phase High Pressure Liquid Chromatography (HPLC) column (SUPELCOSIL LC-18-T, 3 μm particle Size, 15 cm × 4.6 mm, Sigma Aldrich), equilibrated with 100 mM $K_2HPO^4/KH_2PO^4$, 4 mM tetrabutylammonium iodide, pH 5.9, 10% methanol at a flow rate of 1.2 ml $min^{-1}$. cGMP as a reaction product was measured at 260 nm at a retention time of 7 min.

## Spectroscopy

To prepare samples for FTIR, 500 μl of the initial protein solution (1 OD) was concentrated with an Amicon Ultra 10 kDa centrifugal filter to a final OD of ~33 at 540 nm. For deuteration, the protein solution was washed at least five times with deuterium oxide (99%, Sigma) buffer using Amicon Ultra filters and subsequently illuminated using white light to improve intramolecular deuteration. Ten to 15 μl of this solution was then placed on a $BaF_2$ window and concentrated by evaporating solvent water under a stream of dry air. For the CaRGC protein, this step has to be performed with much care to prevent complete dehydration of the sample, since it induces irreversible denaturation of the protein. Samples were then sealed with a second $BaF_2$ window. To ensure reproducible and constant sample thickness, a 6 μm PTFE spacer was placed between the windows. All samples were equilibrated for an hour prior to measurement to minimize temperature drifts. For measurements between 900 and 1800 $cm^{-1}$, an optical cutoff filter at 1850 $cm^{-1}$ was placed in the beamline. The spectral resolution was 2 $cm^{-1}$. Illumination was performed with a pulsed laser for uncaging experiments (330 nm, 6 ns, 10 Hz, 30 mJ per pulse) and time resolved measurements (532 nm). While for continuous illumination a 30 mW continuouswave (CW), a laser with an output maximum of 532 nm was used (no. 37028, Edmund Optics, York, UK). LED illumination was performed with a set of 520 nm LEDs with a FWHM>20 nm. Acquired data was initially processed using OPUS 7.5 software, whereas further processing, including baseline correction with a linear function and pre-spline as well as SVD and global fit procedures, was performed by a customized software developed for Octave 5.1.0.0 initially conceived by Dr. Eglof Ritter.

## Enzyme turnover

All samples were prepared under red light >640 nm. Caged compounds NPE-GTP and NPE-ATP were purchased from Jena Bioscience GmbH (Jena, Germany). For reference, measurements on GTP, cGMP, and PP, as well as the caged ATP and GTP compounds the substrate to $Mn^{2+}$ ratio was 1:2. For measurements of the catalytic activity, the caged compound (NPE-GTP/ATP 50 μl, 10 mM) and manganese (MnCl₂·4H2O 10 μl, 100 mM) were added to the diluted protein solution (400 μl, 1 OD) to ensure sufficient diffusion. After an incubation period of 30 min in the dark, the sample was concentrated with an Amicon Ultra 10 kDa centrifugal filter as described by the manufacturer. Not taking into account the amount of protein-bound substrate the molar ratio in the sample was approximately 1:4:7.4 (RGC:NPE-GTP:$Mn^{2+}$).

## RGC homology model

The CaRGC-43 model was generated using CHARMM33 and PyMol 2.5 based on a homology model of its rhodopsin and linker domains. The crystal structures of the rhodopsin phosphodiesterase from *Salpingoeca rosetta* (SrRhoPDE, PDB-IDs: 7CJ3, 7D7Q) (*Ikuta et al., 2020*) served as templates for

homology modeling using the online platforms for protein structure prediction of Swiss-Modell (*Waterhouse et al., 2018*) and Robetta (*Song et al., 2013*). The crystal waters and the orientation of both protomers were adopted from the template structures. Secondary structure prediction on the full-length *Ca*RGC sequence in JPred4 (*Drozdetskiy et al., 2015*) helped to identify several additional N- and C-terminal features besides the 7-TM-rhodopsin or GC domains as the following: an additional TM-helix 0, an elongated TM-helix 7, short helices on both the N-terminus (helix −1) and the C-terminus (helix 8) and short N-terminal β-sheets. These structures were modeled using CHARMM and then oriented and linked in PyMol in which the cryo-EM maps of the NO-activated human soluble guanylate cyclase (sGC) served as a template (EMDB-ID: EMD-9885) (*Kang et al., 2019*). The final 43-truncated rhodopsin domain was linked to the crystal structure of the guanylate cyclase domain of RGC in complex with GTP (PDB-ID: 6SIR) (*Butryn et al., 2020*) using PyMol.

## Acknowledgements

We are grateful to Christina Schnick and Melanie Meiworm for excellent technical assistance and Joel Kaufmann, Patrick Piwowarski, Anika Spreen and Wayne Busse für critical discussion and suggestions.

The work has been funded by the European Research Council (MERA 693742)(PH), (STARDUST 767092)(PH) and the German Research Foundation (SFB 1078-221545957, B5)(FB), (SFB 1315-327654276)(PH) and Cluster of Excellence Neurocure (EXC2049 390688087)(PH), (EXC2008 390540038)(PH). PH is Hertie Professor for Biophysics and Neuroscience and supported by the Hertie Foundation.

This publication is dedicated to the photoreceptor pioneer Silvia E Braslavksy on the occasion of her 80th birthday.

## Additional information

### Funding

| Funder | Grant reference number | Author |
| --- | --- | --- |
| Deutsche Forschungsgemeinschaft | SFB1315-327654276 | Peter Hegemann |
| Deutsche Forschungsgemeinschaft | EXC2008 390540038 | Peter Hegemann |
| Deutsche Forschungsgemeinschaft | SFB1078 221545957 B5 | Franz Bartl |
| Deutsche Forschungsgemeinschaft | EXC2049 390688087 | Peter Hegemann |
| European Research Council | STARDUST 767092 | Peter Hegemann |
| European Research Council | MERA 693742 | Peter Hegemann |

The funders had no role in study design, data collection and interpretation, or the decision to submit the work for publication.

### Author contributions

Paul Fischer, Conceptualization, Data curation, Formal analysis, Validation, Investigation, Visualization, Methodology, Writing - original draft; Shatanik Mukherjee, Conceptualization, Resources, Supervision, Methodology; Enrico Schiewer, Data curation, Software, Investigation, Visualization; Matthias Broser, Resources, Validation, Investigation, Writing - original draft, Writing - review and editing; Franz Bartl, Resources, Supervision, Funding acquisition, Validation, Methodology, Writing - original draft, Writing - review and editing; Peter Hegemann, Conceptualization, Resources, Funding acquisition, Validation, Methodology, Writing - original draft, Project administration, Writing - review and editing

Author ORCIDs
Paul Fischer (iD) https://orcid.org/0000-0003-3766-9085
Shatanik Mukherjee (iD) http://orcid.org/0000-0002-7359-9339
Enrico Schiewer (iD) http://orcid.org/0000-0001-7913-5597
Franz Bartl (iD) https://orcid.org/0000-0002-0847-867X
Peter Hegemann (iD) https://orcid.org/0000-0003-3589-6452

## Decision letter and Author response
Decision letter https://doi.org/10.7554/eLife.71384.sa1
Author response https://doi.org/10.7554/eLife.71384.sa2

## Additional files
### Supplementary files
- Transparent reporting form

### Data availability
Data files have been provided on Dryad (https://doi.org/10.5061/dryad.6wwpzgmzx) for Figure 4 and 5 as well as for the homology structures of the protein presented in the appendices, upon which Figure 6 is based. This includes the protein prediction by AlphaFold 2.

The following dataset was generated:

| Author(s) | Year | Dataset title | Dataset URL | Database and Identifier |
| --- | --- | --- | --- | --- |
| Fischer P, Mukherjee S, Schiewer E, Broser M, Bartl F, Hegemann P | 2021 | The inner mechanics of rhodopsin guanylyl cyclase during cGMP-formation revealed by real-time FTIR spectroscopy | https://doi.org/10.5061/dryad.6wwpzgmzx | Dryad Digital Repository, 10.5061/dryad.6wwpzgmzx |

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

## Appendix 1

### Experimental setup

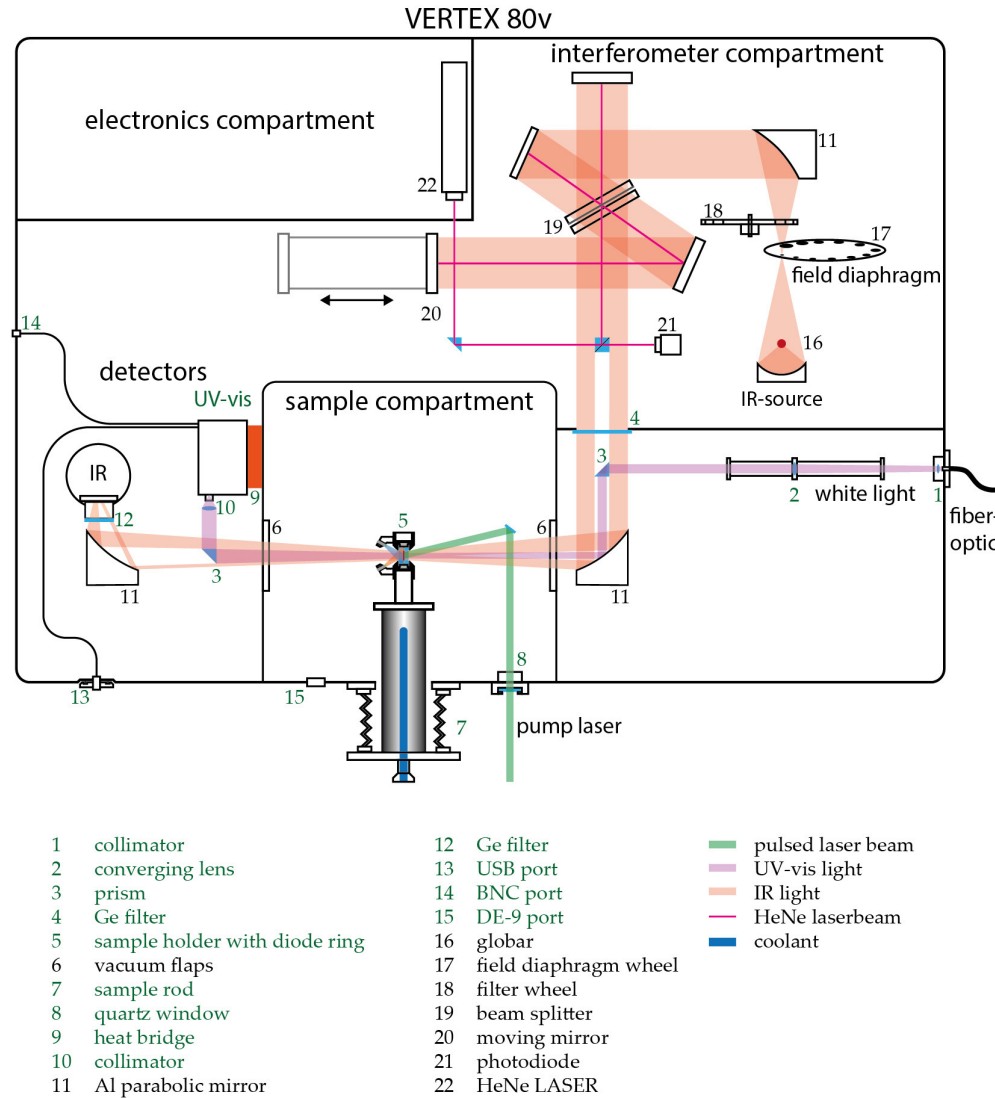

| | | | | |
|---|---|---|---|---|
| 1 | collimator | 12 | Ge filter | ▬ pulsed laser beam |
| 2 | converging lens | 13 | USB port | ▬ UV-vis light |
| 3 | prism | 14 | BNC port | ▬ IR light |
| 4 | Ge filter | 15 | DE-9 port | — HeNe laserbeam |
| 5 | sample holder with diode ring | 16 | globar | ▬ coolant |
| 6 | vacuum flaps | 17 | field diaphragm wheel | |
| 7 | sample rod | 18 | filter wheel | |
| 8 | quartz window | 19 | beam splitter | |
| 9 | heat bridge | 20 | moving mirror | |
| 10 | collimator | 21 | photodiode | |
| 11 | Al parabolic mirror | 22 | HeNe LASER | |

**Appendix 1—figure 1.** Schematic view of the customized Vertex 80 v (Bruker Optics). Custom built parts are indicated green. Depicted numbers are discussed in the Experimental section of the main text.

## Appendix 2

### Uncaging of GTP analog GPCPP

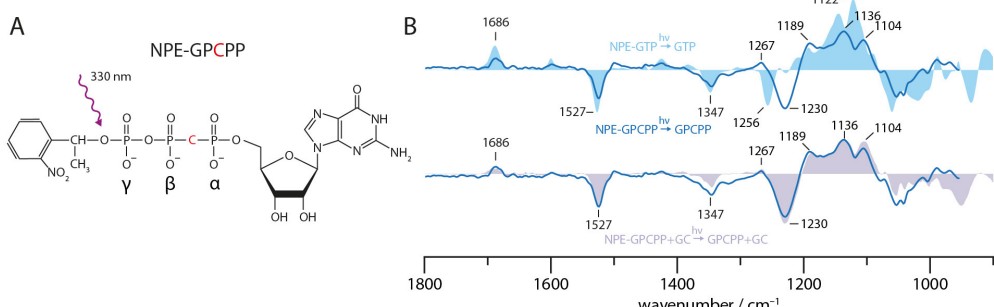

**Appendix 2—figure 1.** Uncaging of NPE-GPCPP. (**A**) Structural formula of NPE-GPCPP. (**B**) UV-light-induced uncaging FTIR difference spectra of free NPE-GPCPP (solid blue line) compared to free NPE-GTP (light blue area) and NPE-GPCPP in presence of GC (light purple area). All spectra were measured in presence of $Mn^{2+}$.

## Appendix 3

### General analysis of CaRGC FTIR spectra

#### amide I

To assign structural changes in the amide I region of *Ca*RGC, H/D exchange has been performed. *Figure 1* shows the L and M state spectra of RGC measured in H2O and D2O. The L state spectrum (A) is acquired at −10°C under single turnover conditions (laser flash, LF) whereas the M state spectrum (B) is obtained via a global fit analysis of the SVD decay components after turning off the 532 nm CW laser (see *Figure 1*).

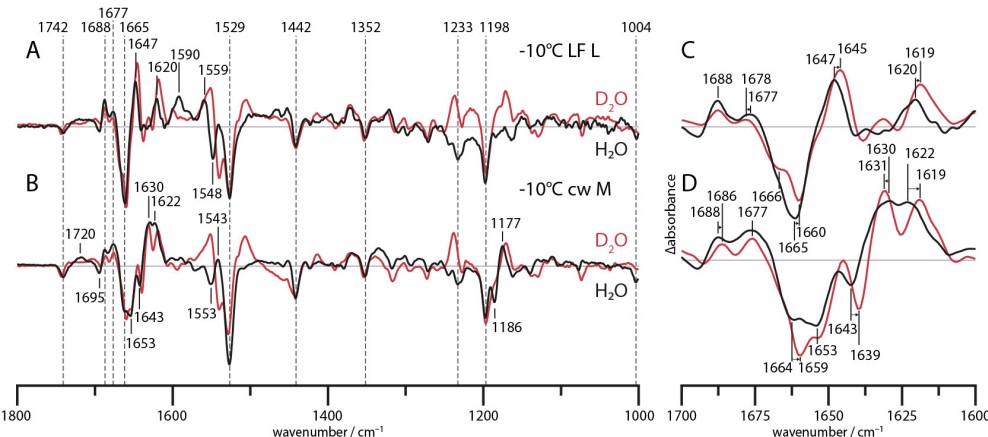

**Appendix 3—figure 1.** L and M state spectra of *Ca*RGC in $H_2O$ (black line) and $D_2O$ (red line). **C** is the magnified amide I region of **A**, measured time resolved after a laser flash at −10°C. **D** shows amide I bands of **B** and has been obtained via a global fit procedure on the SVD decay components after a 1 min illumination with a 532 nm CW laser at −10°C.

In the spectral regime between 1600 and 1700 cm$^{-1}$ (*Figure 1C and D*) the overall band pattern remains roughly conserved upon H/D exchange showing only minor shifts. The strongest displacement is observed for the 1665(-) band which is down-shifted by 5 cm$^{-1}$. This indicates that the bands are most likely caused by structural changes in the protein backbone (*Barth, 2007*). Interestingly, the positive bands at 1688, 1630 and 1622 are typically associated with the formation of β-sheets (*Barth, 2007*; *Goormaghtigh et al., 1994*). E.g., the *Deinococcus Phytochromes* crystal structures show a hairpin, located between the Phytochrome and the GAF domain, forming an antiparallel β-sheet in the Pr state and a α-helical and random coil conformation in the Pfr state (*Burgie et al., 2016*). FTIR studies on similar systems like plant Phytochrome phyA assign the 1630 band to the β-sheet and report a very small H/D shift (<2 cm$^{-1}$) (*Schwinté et al., 2008*). In this case the 1630(+) band depicts no down-shift whatsoever, but rather a miniscule up-shift of around 1 cm$^{-1}$. Due to the broadness of the band the up-shift could only be apparent. While this band is also attributable to other structures, the 1622(+) band is most likely caused by the formation of a β-sheet in the M-state. The speculative interpretation based on the established assignment of amide I bands would be a beginning β-sheet formation in the L state (1688 (+) and 1620 (+)). The 1665(-)/1647(+) double band fits the hydration of a α-helix upon extension in the surrounding cytoplasm (*Mensch et al., 2019*) or water influx as described for channelrhodopsins (*Lórenz-Fonfría et al., 2015*). Upon M state formation, the 1677(+) and 1622(+) bands intensify along with the emerging 1630(+) band which is indicative for the M intermediate. Additionally, two negative bands at 1653(-) and 1643(-) form whereas the former corresponds either to a α-helix or a random coil and the latter most likely to a random coil.

### Chromophore structure (C-C and C=C stretch)

*Figure 1A and B* show the difference spectra of *Ca*RGC corresponding to the L and M intermediates in $H_2O$ (black lines) and $D_2O$ (red lines). One of the most dominant band features are the negative bands at 1529 cm$^{-1}$. They are invariant to H/D exchange allowing an assignment to the retinal

chromophore C=C stretching vibration. The band position corresponds to the spectral absorption of the dark state in the UV-Vis range (450 to 400 nm) (*Neumann-Verhoefen et al., 2013*; *Marcus and Lewis, 1978*). However, no corresponding positive band of the retinal C=C stretching vibration could be identified upon Schiff base deprotonation in the M state spectrum. The spectral region between 1150 and 1250 cm$^{-1}$ is associated with C-C stretching vibrations. Since the C-C stretching modes of the retinal chromophore are typically insensitive to H/D exchange the negative bands at 1198 and 1186 cm$^{-1}$ can be assigned to the retinal chromophore in the dark state. The absence of the 1186 cm$^{-1}$ band in the L state in H$_2$O suggests an overlap of a positive band. The corresponding spectrum in D$_2$O however, shows a positive flank around 1190 cm$^{-1}$ which has been assigned to the C14-C15 stretching vibration of the 13-*cis* retinal (*Mizuide et al., 2006*). In case of rhodopsin phosphodiesterase a most similar pattern (1195(-) and 1183(-)) has been found suggesting all-*trans* to 13-*cis* isomerization as the primary photoreaction (*Watari et al., 2019*) which is typical for microbial rhodopsins (*Ernst et al., 2014*). The same can be assumed for *Ca*RGC.

## Appendix 4

## Secondary structure prediction using JPred4

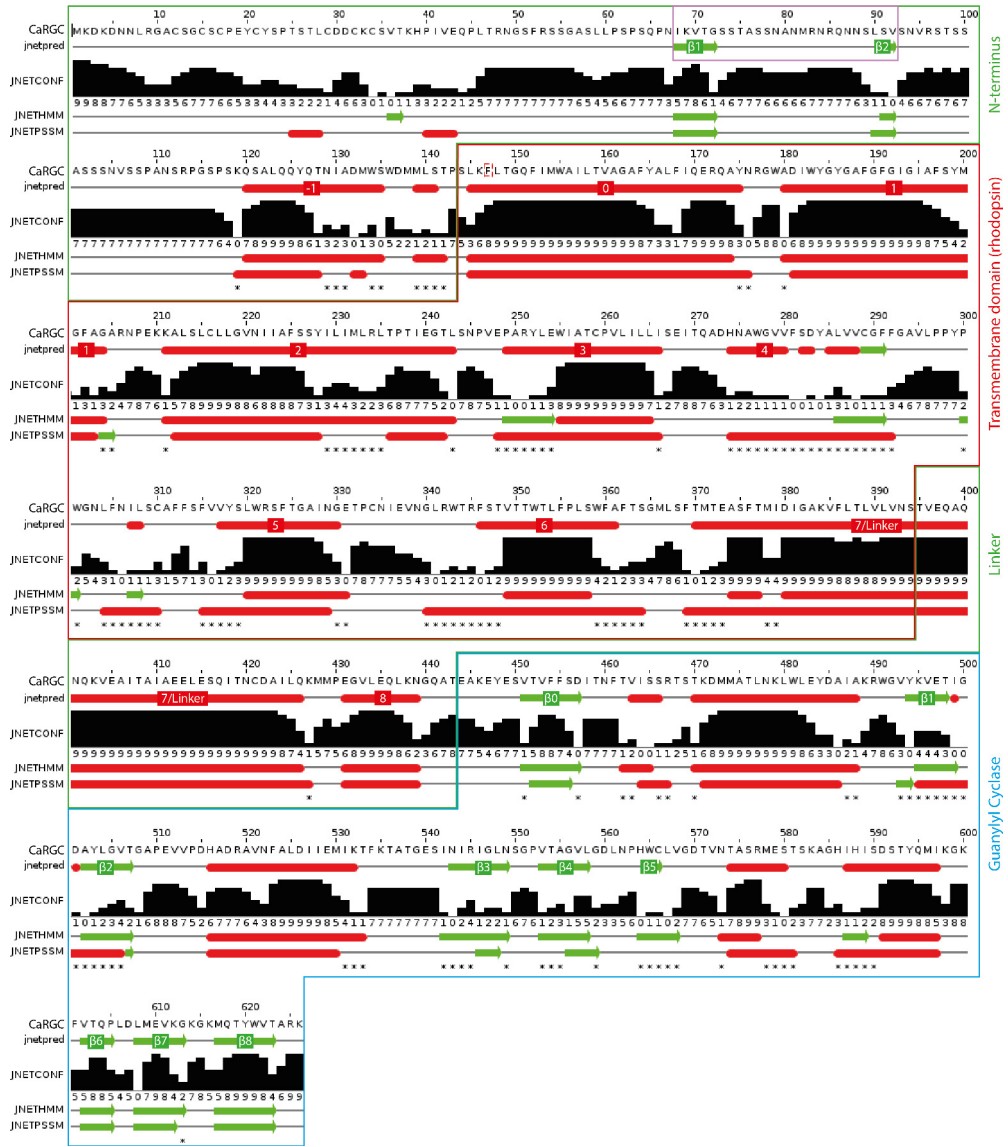

**Appendix 4—figure 1.** Secondary structure prediction for full length *Ca*RGC using JPred434: α-helices are represented as red tubes, β-sheets as green arrows and random coil as grey lines. N-terminal β-sheets β1 and β2 predicted to be involved in cyclase activation are highlighted by a violet frame. Annotations: jnetpred = the consensus prediction. JNETCONF = the confidence estimate for the prediction. High values mean high confidence. JNETHMM - HMM profile based prediction. JNETPSSM = PSSM based prediction. Asterisks are used to rationalize significantly different primary predictions.

# Appendix 5

## Homology model of CaRGC-43

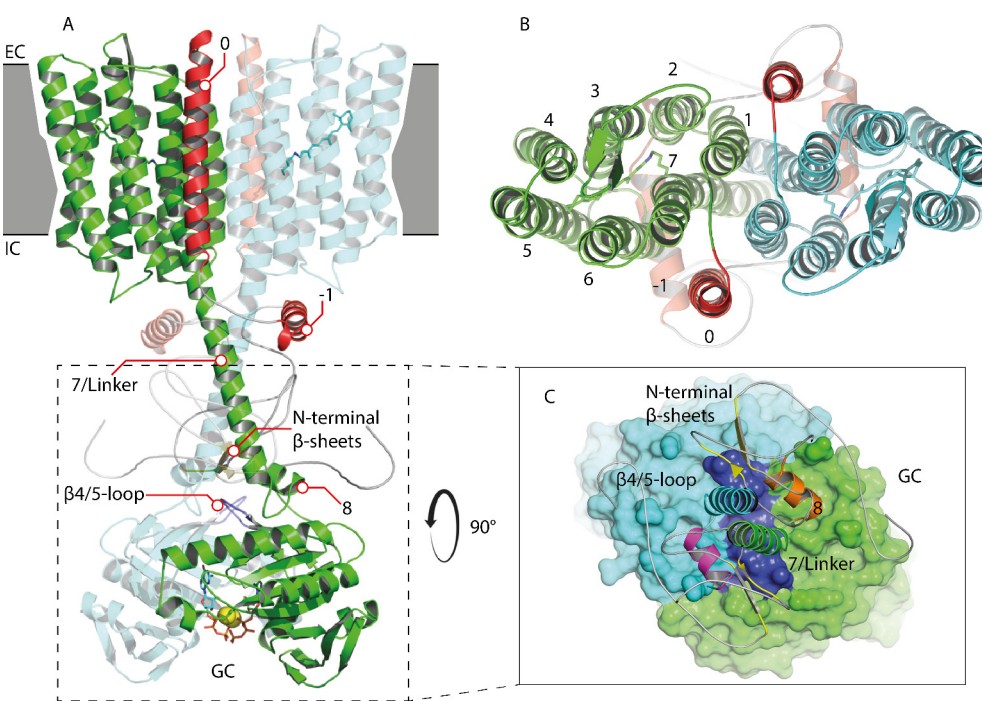

**Appendix 5—figure 1.** Homology model of the 43-truncated CaRGC sequence. (**A**) Cartoon representation of the homology model dimer of CaRGC (lateral view) consisting of the 43-truncated N-terminus, the transmembrane (TM) rhodopsin domain, the elongated helix 7 linker and the GC domain. N-terminal helices 0 and −1 are highlighted in red, N-terminal β-sheets in yellow. The β4/5-loops on the GC top surface are colored in dark blue. Retinal bound to rhodopsin and guanine bound to GC are shown as licorice. $Ca^{2+}$ bound by GC is shown as yellow spheres. (**B**) Top view of the TM domain. Both protomers are colored in green and light blue, respectively. N-terminal helices 0 and −1 are highlighted in red. (**C**) Top view of the GC domain surface including helix 8 and parts of the helix 7 linker and the N-terminus. N-terminal β-sheets are highlighted in yellow. Helix 8 is colored in yellow and orange for better contrast. The β4/5-loops of the GC domain are colored in dark blue.

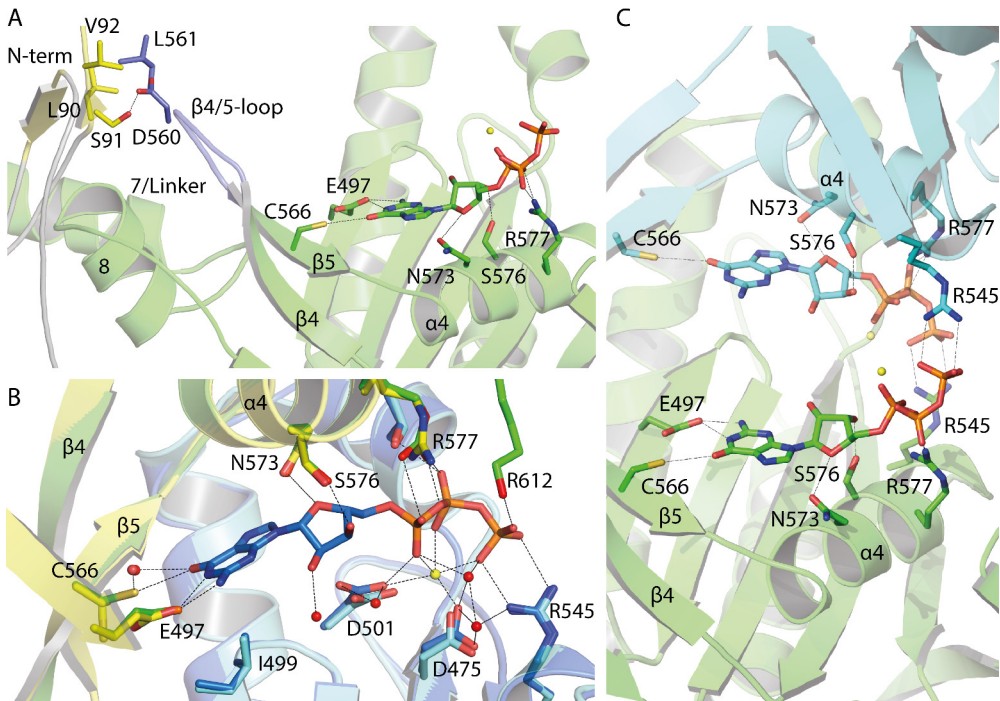

**Appendix 5—figure 2.** Guanine binding pocket of the GC domain and its predicted connection to the *Ca*RGC helix 7 linker and the N-terminus. (**A**) L90, S91 and V92 sidechains of the cytosolic N-terminus are possible interaction partners for L561 and D560 of the GC β4/5-loop, which could trigger the GC activity. The N-terminal β-sheets are represented in yellow and the β4/5-loop in dark blue. Guanine is shown as licorice. $Ca^{2+}$ bound by GC is represented by small yellow spheres. (**B**) Comparison of the guanine binding pocket of the *Ca*RGC homology model and biological assembly 2 of the crystal structure (PDB-ID 6SIR) indicating variations in C566 and R577 side chain orientations. The homology model protomers are colored in green and light blue and the GC crystal structure protomers are colored in yellow and dark blue, respectively. Guanine and the highlighted amino acid sidechains are shown as licorice. Bound $Ca^{2+}$ and close water molecules are represented as yellow and red spheres, respectively. (**C**) Both binding pockets of the *Ca*RGC homology model including guanine substrate and $Ca^{2+}$ ions bound. Both GC protomers are colored in green and light blue, respectively. Guanine and the highlighted amino acid sidechains are shown as licorice. Bound $Ca^{2+}$ is represented as yellow spheres.

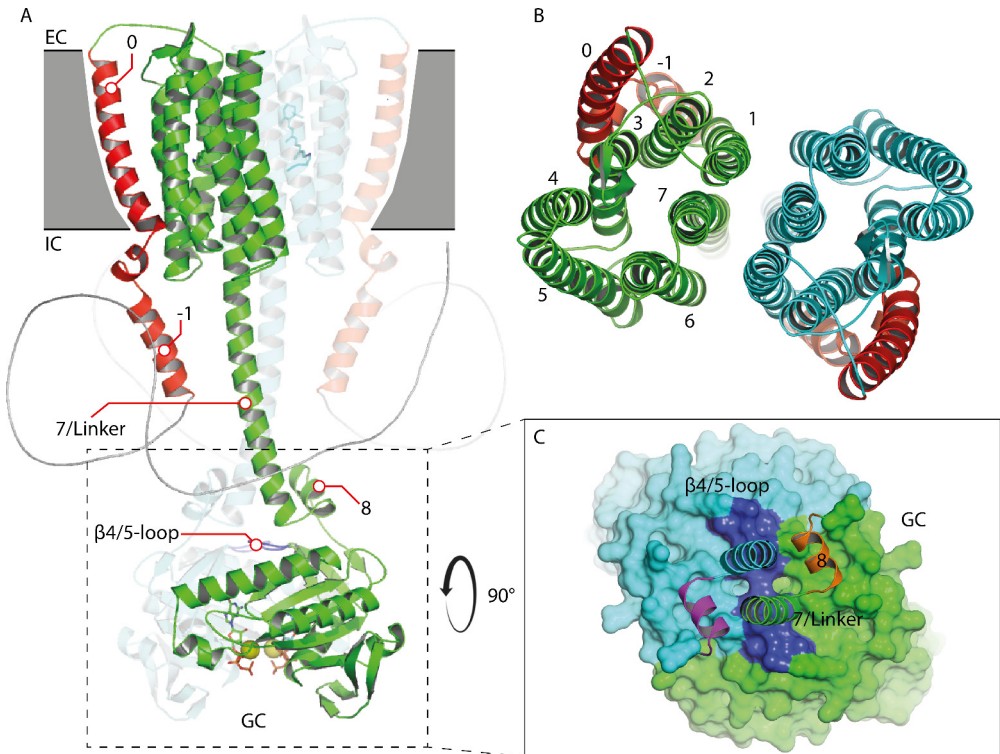

**Appendix 5—figure 3.** Alternative homology model of the 43-truncated *Ca*RGC sequence generated by Alphafold 2 using the *AlphaFold2 advanced* notebook on Google Colab *Mirdita et al., 2021*; *Jumper et al., 2021*. Retinal bound to rhodopsin and guanine and $Ca^{2+}$ bound to GC were added manually using PyMOL 2.5. (**A**) Cartoon representation of the homology model dimer of *Ca*RGC (lateral view). N-terminal helices 0 and −1 (red) show strong deviations from the preceding homology model: helix 0 is located close to helices 2 and 3 comparable to the crystal structure of *Sr*RhPDE (PDB-ID: 7CJ3) (*Ikuta et al., 2020*). Helix -1 takes a rather flexible conformation close to linker helix 7 on the intracellular side. No additional N-terminal features can be found which could influence the GC activity. It has to be noted that it is very unlikely for Alphafold 2 to predict intrinsically disordered structures like most of the N-terminus correctly. Both linker helices 7 are less twisted and helix 8 is almost parallel to helix 7, but their contact site with the GC domain is similar to the preceding homology model. The β4/5-loop is bent towards the GC center compared to the *Ca*GC crystal structure which could be connected to the suppression of enzyme activity. (**B**) Top view of the TM domain. Helices 1 to 7 are oriented similar to the preceding homology model, but the protomers are slightly rotated towards each other: helix 1 is in contact with helix 6 resulting in a larger dimer contact area. (**C**) Top view of the GC domain surface including helix 8 and parts of the helix 7 linker. The GC domain is almost identical to its crystal structure (PDB-ID: 6SIR) (*Butryn et al., 2020*).

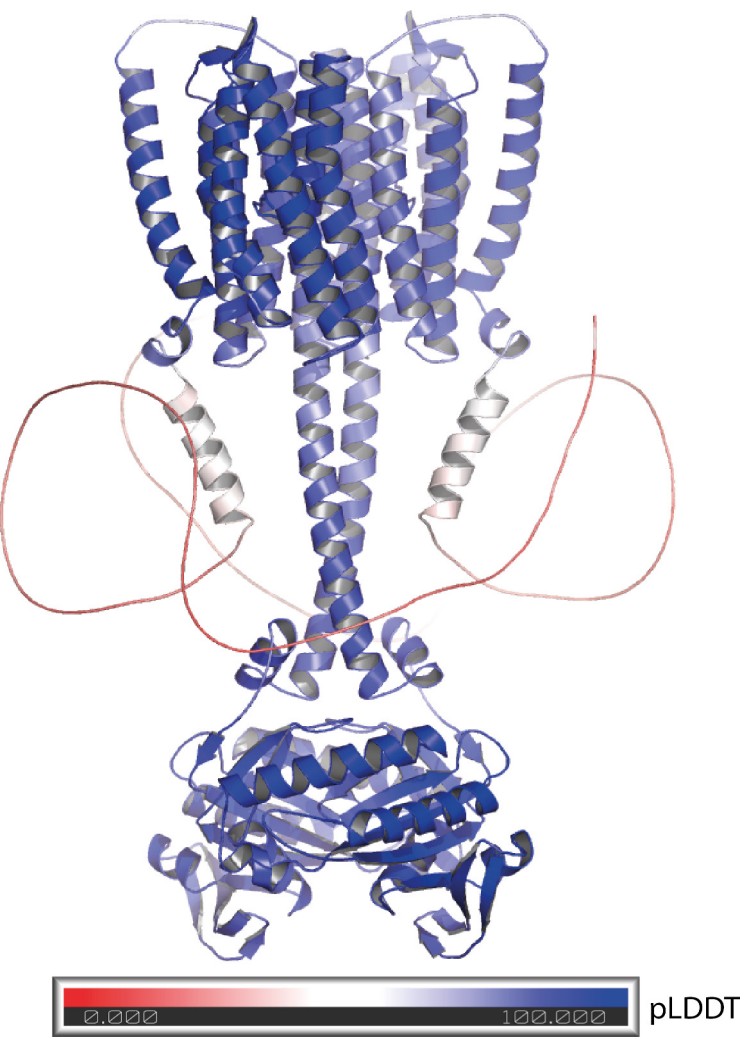

**Appendix 5—figure 4.** Alphafold 2 model of the 43-truncated *Ca*RGC sequence, cartoon representation with color corresponding to the per-residue confidence metric that Alphafold produces (pLDDT *Mariani et al., 2013*). Red to white color (pLDDT < 50) equals low confidence which correlates well with disorder. Blue equals high confidence.

## Appendix 6

## HPLC activity determination

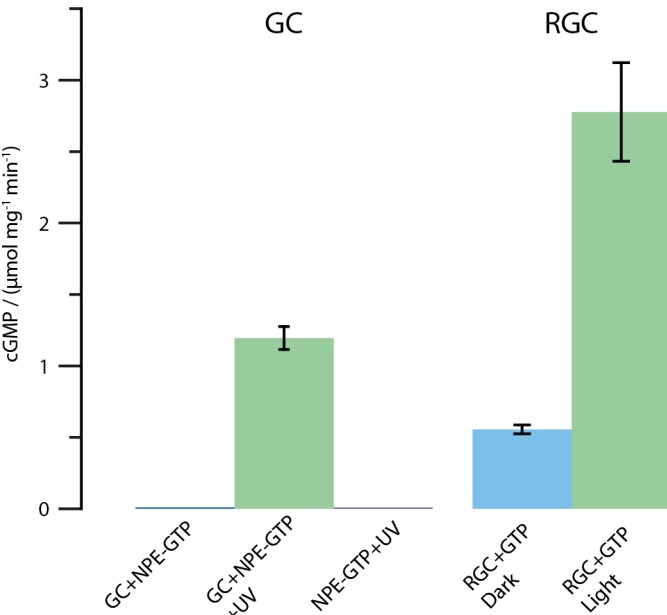

**Appendix 6—figure 1.** Results of the HPLC assay described in the Materials and methods section. Shown is the amount of produced cGMP in μmol per mg protein and minute. All tests were conducted using 2 mM substrate and 2 mM $Mn^{2+}$. The UV illumination was performed with the same protocol as described for the FTIR enzyme turnover experiments (50 pulses at 330 nm, puls duration 6 ns, frequency 10 Hz and 30 mJ per pulse). As shown, free *Ca*GC does not process the caged NPE-GTP substrate unless illuminated with UV light. The full-length *Ca*RGC construct shows an approximately doubled activity compared to free GC when illuminated with green light (532 nm CW laser). The dark activity however, mounts up to roughly 20% in the concentrated detergent samples. Therefore, the caged NPE-GTP construct was used in the FTIR experiments.

