## [Decision Letter]

**Acceptance summary:**

The mechanism of regulation of enzyme activity by rhodopsin domain in Enzymerhodopsins is poorly understood. In this study, Fischer et al., used UV-Vis and FTIR spectroscopy to study the light induced dynamics of rhodopsin guanyl cyclase. Their experiments reveal the role of cytoplasmic N-terminal segments and sheds new light on the mechanics of signal transduction.

**Decision letter after peer review:**

Thank you for submitting your article "The inner mechanics of Rhodopsin Guanylyl Cyclase during cGMP-formation revealed by real-time FTIR spectroscopy" for consideration by *eLife*. Your article has been reviewed by 3 peer reviewers, and the evaluation has been overseen by a Reviewing Editor and Kenton Swartz as the Senior Editor. The following individual involved in review of your submission has agreed to reveal their identity: Hideki Kandori (Reviewer #1).

Essential revisions:

1) The motivation for the general relevance of enzymerhodopsins and in particular RGC should be improved in the introduction

2) The impact of the C259S on the functional cycle compared to wild type could be discussed in more detail.

The seminal work Kuhne et al., PNAS 2019 has to be cited.

3) Figure 2 reports similar spectra for the rhodopsin domain and the full-length protein of CaRGC, from which the authors conclude that the conformation changes of the enzyme domain are too small. This may be true, but I wanted to see the spectra with substrate to confirm that the protein is active. Nevertheless, the authors did not mention at all in the manuscript, and moved to another topic (Figure 3). Then, in the section of "Monitoring enzyme activity by FTIR", the authors used a caged-substrate, and clearly showed the enzyme activity in the sample for FTIR (green spectrum of Figure 4D). Finally the authors concluded the substrate binding in the resting state. If so, why did they use a caged substrate? At that time, I was largely confused and suspicious of their data. Then, in the Discussion section, I found "a substantial RGC dark activity in detergent", which could be the reason of the necessity to use the caged substrate. Materials and methods section includes "HPLC assay for activity determination", but no data are shown in the manuscript. Therefore, I suggest reorganization of the manuscript. In Figure 2, the authors should describe the enzyme activity of this sample that is examined by HPLC. As they use highly concentrated sample (OD ~33), such effect might be taken into account. I think good to show the FTIR data with GTP in Figure 2. If it is difficult because of the dark activity, it should be precisely described in Figure 2.

4) Figure 4 provides the key results of this paper, where molar ratio of enzyme and substrate is not clear. As each concentration can be estimated, the molar ratio should be described in the text.

5) In a further section GC activity in situ is monitored using a caged GTP which is uncaged by a UV laser. FTIR changes are discussed with respect to substrate binding, reorientation of the phosphate residues of GTP and to the in-line attack of the α-phosphate on the 3' OH group of the ribose coordinated by an Asn residue. This is taken to monitor cGMP formation which is likely but please provide more biochemical details and discuss caveats if any.

6) Please address the following concern in your discussion.

The discussion is rather conservative, making few conclusive claims apart from pointing out possibilities how things may proceed. A mechanism of action is then proposed in a model (Figure 6) which predominantly claims that the N-terminal domain of rhodopsin is‚ mechanically' blocking proper assembly of the catalytc dyad. Photoactivation supposedly releases this block and catalysis can proceed. This ignores several earlier observations concerning GC and AC regulation (Ma et al., BMC Struct. Biol. 2010; Vercellino et al., PNAS 2017; Ziegler et al., Febs J. 2017; Qi, C. et al., Science, 2919; Khannpnavar et al., Curr. Opin. Struct. Biol. 2020; Seth et al., Cell. Sign. 2020). In the model in Figure 6 the linker between rhodopsin / receptor and GC effector is most prominent visible, but it is not being functionally discussed. This linker, variably termed ‚helical domain' (Vercellino, Ma, Qi), ‚signaling helix coiled-coil'(Ma), or ‚cyclase-transducing-element' (Ziegler, Seth), is highly conserved in all class III nucleotidyl cyclases. It is implicated in intermolecular signaling in GCs and ACs, possibly as a coiled-coil. It is present in full in the protein MF939579 used in this report. It might be advisable to discuss why this short sequence element with an almost provocative helical domain is discounted in signal transmission in the proposed model. The lack of direct experimental evidence alluding to an involvement is insufficient to exclude participation in signaling.

7) Biochemical GC measurements used an unusual high Mn^2+^ concentration (12 mM). Actual data about GC activity of the full-length protein appear to be missing. Please provide this data.

Perhaps, biochemical activities of other constructs examined by FTIR might be helpful in evaluation of data.

*Reviewer #1:*

Enzymerhodopsins are a recently found new class of rhodopsins, which are composed of transmembrane rhodopsin and cytoplasmic enzyme domains. Light-induced conformational changes in a rhodopsin domain activate an enzyme domain, presumably through domain-domain interaction, but little is known about the molecular mechanism. In this paper, Fischer et al., applied time-resolved UV-visible and FTIR spectroscopy to a rhodopsin guanylyl cyclase from the fungus Catenaria anguillulae (CaRGC). Light-induced spectral changes in the IR region were compared for the rhodopsin domain, CaRGC, and CaRGC with the substrate. They successfully obtained FTIR spectral changes during the catalytic reaction (green spectrum of Figure 4D), which is the first report for enzyme rhodopsins. The experiments are performed in a comprehensive manner and their interpretation is reasonable.

*Reviewer #2:*

Fischer et al., investigate signaling within a multidomain protein using a rhodopsin regulated guanylyl cyclase. FTIR is used almost exclusively as analytic method. This is a sensitive method and experimental details matter. Not all details are outlined and discussed with sufficient clarity (see below). The FTIR data are used as a correlate for conformational changes during signal transduction from the ‚receptor'-rhodopsin modul to the effector GC modul. The emphasis is, therefore, on intermolecular signal transmission from receptor to effector domain, not on the catalytic mechanism of the cyclase reaction.

The authors pay particular attention to known defined states of rhodopsin during the photocycle, thetermed K, L (sub µs to µs states) and M (long-lived state). Live-times of K and L states were mutationally extended by targeted point mutations to make them amenable to FTIR analysis during signaling (excitation by light flashes). In addition, measurements at -10{degree sign}C further slowed the photocycle to make L-states identifiable, the weakly populated K-state could not be unequivocally identified by FTIR. The membrane-bound GC (1-626, a synthetic construct) and several derivatives are expressed in Pichia, solubilized by detergents and purified by Ni-NTA.

The conclusions of this paper are based to a large extent on correlating FTIR data with known structural features of subdomains culminating in an attempted modelling of a signaling pathway.

1) Technically, the actual FTIR measurements appear perfectly executed and a sufficient number of controls validate all data. Difference spectra of the full-length protrein (1-626) are taken to conclude that in the GC domain (443-626) conformational changes are very small and undetectable. This conclusion is in line wih the reported data, yet appears not to be fully in line with published structural data (see below).

2) The discussion is rather conservative, making few conclusive claims apart from pointing out possibilities how things may proceed. A mechanism of action is then proposed in a model (Figure 6) which predominantly claims that the N-terminal domain of rhodopsin is ‚mechanically' blocking proper assembly of the catalytic dyad. Photoactivation supposedly releases this block and catalysis can proceed. This ignores several earlier observations concerning GC and AC regulation (Ma et al., BMC Struct. Biol. 2010; Vercellino et al., PNAS 2017; Ziegler et al., Febs J. 2017; Qi, C. et al., Science, 2919; Khannpnavar et al., Curr. Opin. Struct. Biol. 2020; Seth et al., Cell. Sign. 2020). In the model in Figure 6 the linker between rhodopsin / receptor and GC effector is most prominently visible, but it is not being functionally discussed. This linker, variably termed ‚helical domain' (Vercellino, Ma, Qi), ‚signaling helix coiled-coil'(Ma), or ‚cyclase-transducing-element' (Ziegler, Seth), is highly conserved in all class III nucleotidyl cyclases. It is implicated in intermolecular signaling in GCs and ACs, possibly as a coiled-coil. It is present in full in the protein MF939579 used in this report. It might be advisable to discuss why this short sequence element with an almost provocative helical domain is discounted in signal transmission in the proposed model. The lack of direct experimental evidence alluding to an involvement is insufficient to exclude participation in signaling.

Further to the modelling, more recently progress of structure prediction by AI (or deepmind) has vastly advanced. AlphaFold 2 is now available as open source and on-line (https://alphafold.ebi.ac.uk). One may use alphafold to check on the validity of the presented model. The authors are careful and do not draw firm conclusions from their data and model which be generalized.*Reviewer #3:*

Fischer et al., investigate the molecular structure-activity relationships of a rhodopsin guanylyl cyclase (RGC), which belongs to the class of enzymerhodopsins. They utilize a technically challenging experimental setup to combine UV-Vis and FTIR spectroscopy to study the light-induced dynamics of the catalytic processes. Hydrolysis reactions are triggered using caged compounds. The guanylyl cyclase which is directly linked to the C-terminus is inactive in dark and active illuminated with green light. The chromophore evolution is monitored by UV-Vis spectroscopy and conformational changes of the protein backbone or even individual amino acid residues are followed by IR spectroscopy. The late rhodopsin photoproducts were characterized spectroscopically and it was confirmed that the M state as the active state of the protein. Analysis of truncated variants showed the crucial role of the cytosolic N-terminus in the structural rearrangements upon photo-activation. Also, a regulatory influence of rhodopsin on the substrate binding pocket conformation was shown.

The technically demanding study is carried out carefully and the drawn conclusions are supported by the data. The manuscript is well written and in particular the figures are very descriptive and support nicely the understanding of the experimental workflow and the results.

One potential weak point of the study could be the usage of the C259S rhodopsin variant in order to slow down the photo cycle to allow time scales accessible for measurement.

---

## [Author Response]

Essential revisions:1) The motivation for the general relevance of enzymerhodopsins and in particular RGC should be improved in the introduction

This is a good point; therefore, we emphasized the importance of cGMP and cAMP signaling pathways for biological functions in the introduction. Also, we made its aptitude to complement the optogenetic toolbox clearer and added the following text.

Line 35-42:

“Throughout most branches of life, the ubiquitous intracellular second messenger molecule cGMP is involved in a wide variety of biological functions such as platelet aggregation, neurotransmission, sexual arousal, gut peristalsis, blood pressure, long bone growth, intestinal fluid secretion, lipolysis, phototransduction, cardiac hypertrophy and oocyte maturation [Potter et al., 2011]. Since enzymerhodopsins enable local control of cGMP and, if modified, cAMP [Scheib et al., 2018], they usher in a new direction for optogenetics aiming to unravel the details of these signaling pathways. Thus, a comprehensive understanding of the activation and inactivation of these novel proteins is of utmost interest.”

2) The impact of the C259S on the functional cycle compared to wild type could be discussed in more detail.The seminal work Kuhne et al., PNAS 2019 has to be cited.

We now addressed this important issue in the revised version of the manuscript and the functional role of the C259S mutant for the photocycle is now discussed in more detail. Furthermore, the kinetic constants provided by our experiments are compared to the results of Scheib et al., 2018 and analyzed. Also, possible correlations with ChR2 and parallel 13-*trans*, 15 *anti* and 13 *cis*, 15 *syn* photocycles involving a heterogenic dark state are briefly discussed. In this context, the publication of Kuhne et al., 2019 is cited. We also refer to a publication in progress in which we investigate the DC pair mechanism in CaRGC in detail. Accordingly, we inserted a new text passage:

Line 318-328: “In accordance with our findings, flash photolysis experiments on CaRh WT showed a sequential photoreaction at room temperature [Scheib et al., 2018]. The time constants were reported to be τ_L_=30 ms and τ_M_=570 ms resulting in a 5.5 times retarded M state decay for the C259S mutant (τ_M_=3.13 s) while the L state kinetic remains nearly unaffected (τ_L_=40 ms). Interestingly, the M state spectra of the mutant and the WT (Figure 2) do not exhibit any obvious differences which might hint towards the causal root of the deceleration. A detailed spectroscopic analysis of this phenomenon is currently underway. Despite the clustering of common features, no clear evidence for the population of parallel photocycles starting from 13-*trans*, 15-*anti* or 13-*cis*, 15-*syn* retinal was found for CaRGC as reported for ChR2 [Kuhne et al., 2019]. The slowly increased accumulation of the M state during continuous illumination however (see Figure 1 B), could be an indication of a more complex light adaptation.”

3) Figure 2 reports similar spectra for the rhodopsin domain and the full-length protein of CaRGC, from which the authors conclude that the conformation changes of the enzyme domain are too small. This may be true, but I wanted to see the spectra with substrate to confirm that the protein is active. Nevertheless, the authors did not mention at all in the manuscript, and moved to another topic (Figure 3). Then, in the section of "Monitoring enzyme activity by FTIR", the authors used a caged-substrate, and clearly showed the enzyme activity in the sample for FTIR (green spectrum of Figure 4D). Finally the authors concluded the substrate binding in the resting state. If so, why did they use a caged substrate? At that time, I was largely confused and suspicious of their data. Then, in the Discussion section, I found "a substantial RGC dark activity in detergent", which could be the reason of the necessity to use the caged substrate. Materials and methods section includes "HPLC assay for activity determination", but no data are shown in the manuscript. Therefore, I suggest reorganization of the manuscript. In Figure 2, the authors should describe the enzyme activity of this sample that is examined by HPLC. As they use highly concentrated sample (OD ~33), such effect might be taken into account. I think good to show the FTIR data with GTP in Figure 2. If it is difficult because of the dark activity, it should be precisely described in Figure 2.

We agree with this point and thank the reviewer to address this important question. The necessity to use caged substrate due to the high observed dark activity is now described in the context of Figure 2. The FTIR experiments with uncaged GTP were conducted several times but never indicated any substrate turnover. We attribute this to the substantial dark activity which led to substrate depletion during the sample preparation and equilibration (~1 h). We decided not to show the spectra of the full-length protein with substrate since they do not deviate from the depicted RGC spectra in Figure 2 and do not provide any additional information.

We also provided the results of the HPLC analysis to the appendix in a new figure (Appendix 6), which demonstrates substantial dark activity of RGC in detergent. As a potential enhancer, the high protein concentration is mentioned. Please note, that OD 33 only refers to the “regular” FTIR spectra. Experiments related to enzyme activity and substrate turnover are conducted with a lower concentration (~0.3 mM). We now added in Line 139-143:

“Experiments on the full-length RGC with GTP substrate did not reveal any differences and were therefore not shown. This apparent lack of enzyme activity is attributable to the dark activity observed in the HPLC activity measurements which were conducted as control (Appendix 6). During equilibration in the spectrometer, the available GTP has already been turned over and is therefore unobservable in the difference spectrum.”

and Line 181-185: “To verify enzymatic activity, protein of all samples was also tested via an HPLC assay (Appendix 6) which demonstrated on the one hand the necessity to use a photolabile caged GTP substitute which is not turned over by the RGC dark activity and on the other hand, that cGMP is successfully produced by free GC when illuminating the caged NPE-GTP compound with UV-light.”

and Line 407-408: “or the high protein concentration in detergent”

4) Figure 4 provides the key results of this paper, where molar ratio of enzyme and substrate is not clear. As each concentration can be estimated, the molar ratio should be described in the text.

We agree with the reviewer and estimated the molar ratio. This information is now provided in the Methods and Materials section Line 632-633:

“Not taking into account the amount of protein bound substrate the molar ratio in the sample was approximately 1:4:7.4 (RGC:NPE-GTP:Mn^2+^)”

5) In a further section GC activity in situ is monitored using a caged GTP which is uncaged by a UV laser. FTIR changes are discussed with respect to substrate binding, reorientation of the phosphate residues of GTP and to the in-line attack of the α-phosphate on the 3' OH group of the ribose coordinated by an Asn residue. This is taken to monitor cGMP formation which is likely but please provide more biochemical details and discuss caveats if any.

To verify the enzymatic activity, protein used in the FTIR experiments was also analyzed via an HPLC assay which is now described in detail in the “Materials and methods” section and the most relevant results were added to the Appendix 6. The latter are also mentioned in the Results section to emphasize the necessity to use caged compounds and to verify the turnover of GTP to cGMP. This is in great accordance to the cGMP absolute signal shown in Figure 5 A which fits the emerging band pattern in Figure 4 D (green line) indicating GTP to cGMP turnover by RGC. Please also see changes under point 3). We added at line 595-603:

“For the activity assays of free GC, 1 µM protein was mixed with 1 mM GTP or NPE-GTP and 1 mM Mn^2+^ in a final volume of 100 µl. The incubation time was 10 minutes. Enzymatic activity was stopped by flash freezing in liquid nitrogen and addition of 200 µl 0.1 N HCl. After centrifugation (90 s, 12,000×g, RT), the supernatant was filtered through a 0.2 µm chromafil filter (Macherey-Nagel) and 25 µl was applied on a C18 Reversed Phase High Pressure Liquid Chromatography (HPLC) column (SUPELCOSIL LC-18-T, 3 µm particle Size, 15 cm × 4.6 mm, Σ Aldrich), equilibrated with 100 mM K_2_HPO_4_,/KH_2_PO_4_, 4 mM tetrabutylammonium iodide, pH 5.9, 10 % methanol at a flow rate of 1.2 ml min^-1^. cGMP as a reaction product was measured at 260 nm at a retention time of 7 minutes.”

6) Please address the following concern in your discussion.The discussion is rather conservative, making few conclusive claims apart from pointing out possibilities how things may proceed. A mechanism of action is then proposed in a model (Figure 6) which predominantly claims that the N-terminal domain of rhodopsin is ‚mechanically' blocking proper assembly of the catalytc dyad. Photoactivation supposedly releases this block and catalysis can proceed. This ignores several earlier observations concerning GC and AC regulation (Ma et al., BMC Struct. Biol. 2010; Vercellino et al., PNAS 2017; Ziegler et al., Febs J. 2017; Qi, C. et al., Science, 2919; Khannpnavar et al., Curr. Opin. Struct. Biol. 2020; Seth et al., Cell. Sign. 2020). In the model in Figure 6 the linker between rhodopsin / receptor and GC effector is most prominent visible, but it is not being functionally discussed. This linker, variably termed ‚helical domain' (Vercellino, Ma, Qi), ‚signaling helix coiled-coil'(Ma), or ‚cyclase-transducing-element' (Ziegler, Seth), is highly conserved in all class III nucleotidyl cyclases. It is implicated in intermolecular signaling in GCs and ACs, possibly as a coiled-coil. It is present in full in the protein MF939579 used in this report. It might be advisable to discuss why this short sequence element with an almost provocative helical domain is discounted in signal transmission in the proposed model. The lack of direct experimental evidence alluding to an involvement is insufficient to exclude participation in signaling.

We thank the reviewer for this suggestion which gives us the opportunity the discuss the mechanism in more detail and in a more general context. The linker domain has been found to enable GC and AC regulation in other constructs. Also, the structure of the linker shows great resemblance to the S-helical domains responsible for regulation in these proteins. This is now discussed and the possibility of a concerted action of N-terminus and linker is mentioned. The mentioned publications are cited in this context. Also, a second structural model has been created with AlphaFold2 showing an alternative arrangement of the N-terminus. The activation mechanism depicted in Figure 6 was adapted accordingly and the possibility for an enzyme regulation via the linker as well as a potential interaction between linker and the N-terminal helix -1 is discussed. To address this point we added Line 372-384:

“The functional involvement of the C-terminal linker domain however, could not be observed in the FTIR experiments. Nonetheless, a signal propagation via the linker domain including the shorter handle helices at the GC surface cannot be excluded. Structural alterations in this segment could preserve the peptide environment, thus rendering a possible movement unobservable by the method. Slight rotations or a scissor like movement in a constantly hydrated environment would not precipitate strong amide mode changes. Smaller amide features however, could be masked behind more dominant band patterns of the Rh segment. Since highly conserved linker constructs are involved in GC and AC regulation [Ma2010, Vercellino2017, Ziegler2017, Qi2019, Khannpnavar2020, Seth2020], a similar mechanism cannot be disregarded to play a role in RGCs as well. The *Ca*RGC transducer element is predicted to form a structural motif consisting of an elongated TM and a short handle helix which is most similar to the S-helical domains found in e.g. Gsα. Therefore, signal propagation presumably requires a concerted reorientation of both, linker domain and N-terminus.”

Line 479-481: “Since an involvement of the C-terminal linker helices in the photoactivation could not be observed, potential alterations in this segment are not highlighted in the model. Nevertheless, a concerted action of linker and N-terminus appears possible.”

7) Biochemical GC measurements used an unusual high Mn^2+^ concentration (12 mM). Actual data about GC activity of the full-length protein appear to be missing. Please provide this data.Perhaps, biochemical activities of other constructs examined by FTIR might be helpful in evaluation of data.

Thank you for detecting this! The 12 mM concentration was a typo for which we apologize. We conducted these experiments with a Mn^2+^ concentration of 2 mM which is in accordance to activity studies on the construct reported by Scheib et al., 2018. This error was corrected.